# Systematic identification of post-transcriptional regulatory modules

Matvei Khoroshkin[1,2,3,4,14], Andrey Buyan [5,14], Martin Dodel [6,7,14], Albertas Navickas [1,2,3,4,13], Johnny Yu[1,2,3,4], Fathima Trejo[8], Anthony Doty[8], Rithvik Baratam[1,2,3,4], Shaopu Zhou[1,2,3,4], Sean B. Lee[1,2,3,4], Tanvi Joshi[1,2,3,4], Kristle Garcia[1,2,3,4], Benedict Choi[1,2,3,4], Sohit Miglani[1,2,3,4], Vishvak Subramanyam[1,2,3,4], Hailey Modi[9,10,11], Christopher Carpenter[1,2,3,4], Daniel Markett[1,2,3,4], M. Ryan Corces [9,10,11], Faraz K. Mardakheh [6,7] ✉, Ivan V. Kulakovskiy [5,12] ✉ & Hani Goodarzi [1,2,3,4] ✉

In our cells, a limited number of RNA binding proteins (RBPs) are responsible for all aspects of RNA metabolism across the entire transcriptome. To accomplish this, RBPs form regulatory units that act on specific target regulons. However, the landscape of RBP combinatorial interactions remains poorly explored. Here, we perform a systematic annotation of RBP combinatorial interactions via multimodal data integration. We build a large-scale map of RBP protein neighborhoods by generating in vivo proximity-dependent biotinylation datasets of 50 human RBPs. In parallel, we use CRISPR interference with single-cell readout to capture transcriptomic changes upon RBP knockdowns. By combining these physical and functional interaction readouts, along with the atlas of RBP mRNA targets from eCLIP assays, we generate an integrated map of functional RBP interactions. We then use this map to match RBPs to their context-specific functions and validate the predicted functions biochemically for four RBPs. This study provides a detailed map of RBP interactions and deconvolves them into distinct regulatory modules with annotated functions and target regulons. This multimodal and integrative framework provides a principled approach for studying post-transcriptional regulatory processes and enriches our understanding of their underlying mechanisms.

RNA binding proteins (RBPs) are crucial for governing all stages of post-transcriptional regulation, from RNA splicing and nuclear export to translation and decay. Despite the limited number of conventional RBPs encoded in the human genome (fewer than 1500)[1], they shepherd more than 100,000 transcripts throughout their life cycles. Therefore, it is unlikely that any given RBP acts on only one specific regulon – defined as a group of transcripts that are regulated as a unit through the same regulatory factors[2–4] – or performs only one specific function.

[1]Department of Biochemistry and Biophysics, University of California, San Francisco, San Francisco, CA, USA. [2]Department of Urology, University of California, San Francisco, San Francisco, CA, USA. [3]Helen Diller Family Comprehensive Cancer Center, University of California, San Francisco, San Francisco, CA, USA. [4]Bakar Computational Health Sciences Institute, University of California, San Francisco, San Francisco, CA, USA. [5]Institute of Protein Research, Russian Academy of Sciences, Pushchino, Russia. [6]Centre for Cancer Cell and Molecular Biology, Barts Cancer Institute, Queen Mary University of London, London, UK. [7]Department of Biochemistry, University of Oxford, Oxford, UK. [8]College of Arts and Sciences, University of San Francisco, San Francisco, CA, USA. [9]Gladstone Institute of Neurological Disease, San Francisco, CA, USA. [10]Gladstone Institute of Data Science and Biotechnology, San Francisco, CA, USA. [11]Department of Neurology, University of California San Francisco, San Francisco, CA, USA. [12]Vavilov Institute of General Genetics, Russian Academy of Sciences, Moscow, Russia. [13]Present address: Institut Curie, UMR3348 CNRS, Inserm, Orsay, France. [14]These authors contributed equally: Matvei Khoroshkin, Andrey Buyan, Martin Dodel. ✉e-mail: faraz.mardakheh@bioch.ox.ac.uk; ivan.kulakovskiy@gmail.com; hani.goodarzi@arcinstitute.org

Instead, RBPs assemble into units of post-transcriptional control in a combinatorial manner to cover a wide array of functions for thousands of distinct target regulons[5]. Similarly, the set of transcripts bound by a given RBP does not represent a single regulon; instead, these transcripts are part of various independent regulons, each characterized by the distinct set of RBPs that act on it. This combinatorial RBP interaction network enables a limited number of RBPs to fulfill diverse roles in post-transcriptional regulation, governing all aspects of the life cycle for all RNAs in the cell. This complexity illustrates the need to systematically understand the regulatory context and consequence of each RBP in relation to each target transcript and their associated regulons.

Several recent large-scale projects[6], such as the work by the ENCODE consortium, have focused on mapping the interactions between RBPs and their binding partners[7]. Other studies have explored the subcellular localization of hundreds of RBPs and RNAs, and the gene expression changes that result from RBP knockdowns[8–10]. However, these transcriptome-wide maps of RBP-RNA interactions have yet to fully elucidate the specific regulatory consequences of each binding event. Unlike the clearer picture in transcriptional regulation, where transcription factor binding at several target loci often implies their co-regulation, the inherent diversity in post-transcriptional regulatory processes, from processing to decay, suggests that RBPs interact with specific RNAs as parts of distinct regulatory networks, leading to a range of possible functional outcomes[11,12]. Recognizing this gap, our study aims to move beyond mapping the RBP interactome[13] to define "functional regulatory modules": groups of RBPs that closely interact, either physically or functionally, to regulate specific sets of transcripts defining each target regulon.

Delineating regulatory modules is challenging due to the multi-faceted nature of interactions between RBPs, which extend beyond simple physical associations. RBPs can co-localize, directly interact, or cooperate by binding to the same RNAs, either simultaneously or sequentially[14]. To capture this complexity, here we adopted a multi-modal approach to develop an Integrated Regulatory Interaction Map (IRIM) that integrates three types of functional interactions: (i) physical co-localization, (ii) binding to the same RNA targets at varying times and locations, and (iii) participation in the same regulatory pathway leading to similar transcriptomic changes. The latter serves as our method to capture genetic interactions (GIs), which provide a nuanced view of gene functions by capturing complex, context-dependent interplays between genes[15,16]. To showcase the utility of our framework, we further explored several regulatory roles predicted by our approach for specific RBPs. In particular, we experimentally validated that two RBPs, ZC3H11A and TAF15, both control independent regulons through distinct regulatory programs that include regulation of alternative splicing, RNA translation, or stability, depending on the regulon. Our findings also highlighted several RBPs, such as ZNF800 and QKI, that are involved in both transcriptional and post-transcriptional gene expression regulation, emphasizing the complexity of RBP action. Taken together, this study provides a systematic and principled approach that enhances our understanding of the complex and multifaceted roles of RBP functional modules in gene regulation.

## Results

### Integrated RBP interaction maps to reveal regulatory modules

In order to broadly and systematically annotate regulatory interactions between RBPs, we combined data from three independent modalities, namely (i) RBP-RBP physical associations revealed by BioID2-mediated proximity protein labeling, (ii) RBP-RBP genetic associations identified through Perturb-seq, and (iii) RBP-RNA interactomes extracted from the ENCODE eCLIP dataset (Fig. 1 and Supplementary Fig. S1). First, we fused the BioID2 system to 50 RBPs in K562 cells, validated expression of each fusion by western blotting, and captured the protein

neighborhood of each RBP using streptavidin pulldown and mass spectrometry[17]. Importantly, for each RBP, we also included matched controls by processing the same lines without a biotin pulse, which was crucial for generating a high-confidence protein neighborhood dataset to systematically identify co-localizing RBPs (Supplementary Data Files 3, 6, 7). Second, we used Perturb-seq, a parallelized loss-of-function screen with rich single-cell transcriptomic readouts[15], to reveal sets of RBPs whose perturbations similarly impact the gene expression landscape of the cell (Supplementary Fig. S5). We obtained transcriptome-wide gene expression measurements following depletion of 68 RBPs representing a variety of regulatory processes (see "Methods") (Supplementary Data File 5), and used the resulting high-dimensional data to systematically delineate genetic interactions between RBPs[15]. Finally, we re-analyzed the ENCODE eCLIP dataset to evaluate the extent to which pairs of RBPs bind to common RNA targets[18].

The abovementioned data modalities capture complementary aspects of regulatory interactions between RBPs. Therefore, integrating these sources of information is a critical step toward generating a more comprehensive and generalizable map of regulatory interactions (Fig. 1 and Supplementary Fig. S1). To accomplish this, we first generated RBP-target interaction maps for each individual modality, where the 'target' can be either neighboring protein (BioID), downstream gene (Perturb-seq), or target RNA (eCLIP; Supplementary Fig. S2A–C). In order to make the measurements comparable between datasets, we standardized them across the target features. We posited that RBPs that fall close to each other in this feature space, which reflects physical and functional proximity, function as part of the same regulatory modules. Therefore, for each data modality, we estimated pairwise cosine distances between RBPs and transformed them into empirical $p$-values to achieve a uniform scale for pairwise similarity. Finally, the three separately calculated $p$-values for RBP-RBP similarities (i.e., from BioID, Perturb-seq, and eCLIP, respectively) were combined into a single unified probability score (Supplementary Fig. S1) expressing the overall likelihood of functional interactions between pairs of RBPs (Supplementary Data File 1).

The resulting 'Integrated Regulatory Interaction Map' (IRIM) provides the means to elucidate the combinatorial regulatory logic underlying RBP-mediated post-transcriptional control of gene expression (Fig. 2A). Interaction maps are often interpreted by identifying proteins that cluster together into functional complexes in an unsupervised manner. As shown in Fig. 2A, IRIM similarly captures a number of canonical RBP modules involved in key post-transcriptional regulatory programs such as 'cytoplasmic translation' and 'splicing'. However, we also observe many off-diagonal interactions in IRIM that are indicative of RBPs with multiple functions in different aspects of RNA regulation. Moreover, IRIM captures 20% more regulatory interactions than an analogous map built based on the current state-of-the-art protein-protein interaction database, STRING-DB[19], which also incorporates indirect (functional) associations (Supplementary Fig. S2D).

Our integrative approach brings together RBPs that form key regulatory modules – which we define as a set of RBPs that share significant functional interactions (Supplementary Data 16) – and broadly recapitulates what is known about the functions of these RBPs. It also allows us to delineate regulons associated with each regulatory module, which we define as the set of RNA targets that bind at least 2 RBPs participating in the module according to the eCLIP binding data (Supplementary Data 18). However, tracking the source of the signal to the individual input modalities is also often informative, which is readily doable with our setup. For example, among the group of 15 RBPs that are collectively associated with ribosome biogenesis and translation-related processes, RBPs such as FXR1, ZNF622, and ZNF800 bind overlapping RNA targets based on the eCLIP data, whereas UCHL5 and AGGF1, which have been previously shown to

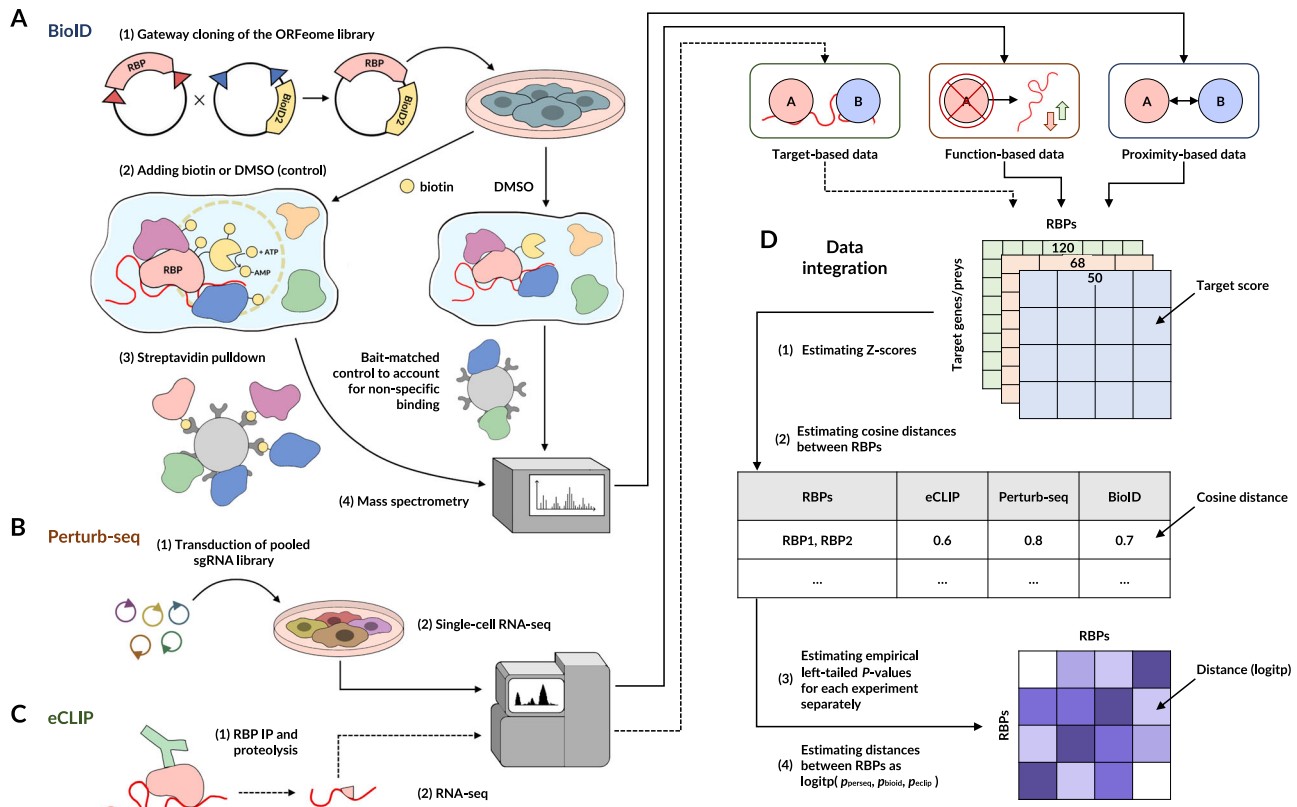

**Fig. 1 | Workflow overview: generating an integrated regulatory interaction map of RNA-binding proteins. A–C** The results of BioID2, Perturb-seq, and publicly available ENCODE eCLIP assays were independently processed and normalized across RBPs. **D** The resulting Z-scores were used to estimate the cosine distance between all pairs of the tested RBPs and to calculate empirical left-tailed *p*-values for RBP-RBP similarities. For each pair of RBPs, the *p*-values from three assays were aggregated as in ref. 117 to obtain a single measure of similarity between RBPs across the feature spaces from the three modalities. The resulting matrix of pairwise similarities was defined as the Integrated Regulatory Interaction Map (IRIM) that simultaneously captures physical and functional interactions between RBPs.

inhibit p53 ubiquitination by MDM2[20,21], are additionally associated with the regulation of p53-mediated apoptosis based on Perturb-seq results (Fig. 2A). Another example is the group of RBPs associated with mitochondrial and cytoplasmic RNA metabolism; while eCLIP data brings together the RBPs that tend to bind the same RNA classes, proximity labeling clearly distinguishes mitochondrial RBPs (TBRG4, FASTKD2, and SUPV3L1) from others (Fig. 2A).

Having systematically revealed inter-RBP interactions, encompassing both known and potentially novel associations, we next set out to confirm that the identified interactions align with established, "gold-standard" databases. For this, we matched our findings, derived from IRIM, against interactions cataloged in STRING[19], OpenCell[5], hu.MAP[22] and Zanzoni et al.[23]. Permutation tests revealed a statistically significant overlap between our detected interactions and these databases (FDR = 0.031 for STRING, 0.00017 for OpenCell, 0.01 for hu.MAP and 0.0015 for Zanzoni et al. using a 0.25 quantile as the integrated distance threshold) (Fig. 2F and Supplementary Fig. S3A). At this threshold, we identified 1001 RBP-RBP pairs, with 776 of these interactions being novel (not reported in the STRING database), and an average of 22 contacts per RBP, a five-fold increase in interactions compared to STRING (see Supplementary Data File 14). Moreover, the newly reported interactions that were not annotated in STRING showed significant intersection with OpenCell inter-RBP interactions ($p < 10^{-4}$), supporting the validity of the newly reported interactions. IRIM remained robust to the removal of any one data modality, maintaining a significant intersection with STRING (Supplementary Fig. S3A, see "Methods"). This alignment with established databases validates our approach, emphasizing its effectiveness in revealing novel, meaningful RBP interactions.

To further assess the robustness of IRIM and the resulting RBP associations, we performed a randomization test, permuting various percentages (5, 10, 25, 50, 75, and 100%) of the matrix columns, each column representing the distances from a given RBP to all the other RBPs (Supplementary Fig. S3B). To ensure that our set of selected RBPs sufficiently represent annotated RNA-binding proteins, we also compared the resilience of IRIM's topology to that of STRING-DB, OpenCell and hu.MAP by systematically introducing noise to these datasets (Fig. 2E). The similar rates of degradation in IRIM and STRING under increasing noise regimens highlight the ability of the selected RBPs to maintain the overall network topology; even at 25% of data randomly altered, the ranking of RBP neighbors remains largely unchanged. This persistent stability underscores a resilient modular structure in IRIM, reinforcing the resilience of our delineated interactions to the addition or removal of other RBPs.

## Combinatorial interactions between RBPs provide a molecular basis for their multifaceted role in gene regulation
IRIM reveals numerous cases of combinatorial interactions and functionally pleiotropic roles for RBPs, a number of which have been previously described. Such combinatorial interactions appear as off-diagonal groupings in IRIM (Fig. 2A). For example, IRIM shows that both U2AF1 and KHSRP associate with clusters related to splicing and translation (Fig. 2A); direct roles for these RBPs in regulating both these processes have been recently reported[24,25]. In addition, functionally pleiotropic RBP groups are assigned to multiple clusters using the fuzzy clustering c-means approach, revealing the interconnections between rRNA transcription, splicing, and translational processes (Supplementary Fig. S3C). Furthermore, we performed a

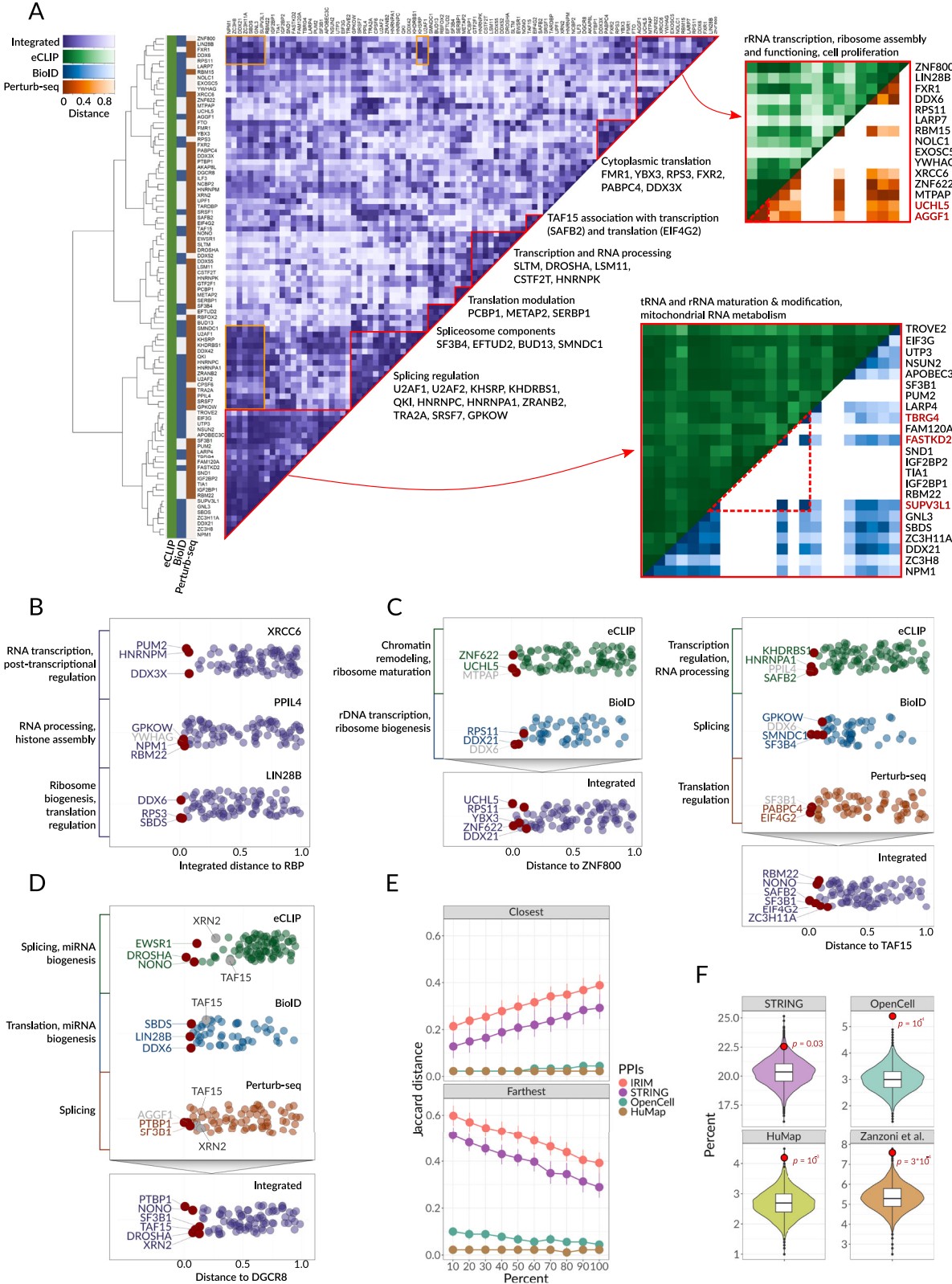

graph-based visualization of the IRIM interactions to demonstrate the identified RBP functional clusters (Supplementary Fig. S3D).

To go beyond known examples and to gain insights into previously unknown functions of RBPs, we implemented a label transfer approach for each RBP to extrapolate annotated functions of closest neighbors in IRIM to infer possible functions of the RBP of interest. As expected, we find that the closest neighbors often capture the known

functions of RBPs (Fig. 2B and Supplementary Fig. S4). For instance, interactions of the non-homologous end joining effector XRCC6 (Ku70) and key RNA regulatory proteins PUM2 (translational repression), HNRNPM (mRNA processing), and DDX3X (RNA helicase) hint at a connection between RNA metabolism and the DNA damage response, with supporting studies showing PUM2 driving chromosomal instability and DDX3X colocalizing with double strand breaks[26,27].

**Fig. 2 | Unveiling post-transcriptional regulatory modules through integrative analysis of RBP-RBP interactions. A** Integrated Regulatory Interaction Map (IRIM): This heatmap displays integrated distances between RBPs, where each cell's color denotes the integrated distance between the corresponding RBPs. Hierarchical clustering is illustrated by the dendrogram to the left. The colormap signifies the inclusion of RBPs in three data sources: eCLIP (green), BioID (blue), and Perturb-seq (brown). Recognized regulatory modules are emphasized in red with contributing RBPs labeled directly on the plot. Insets present detailed heatmaps for two exemplary modules, colored respectively for source datasets: BioID (blue), Perturb-seq (orange), and eCLIP (green). Proteins discussed are highlighted in red, and examples of module interplay, including U2AF1 and KHSRP, are marked in orange. Source data are provided as a Source Data file. **B** Swarm Plots for RBP Partners of XRCC6, PPIL4, and LIN28B: Swarm plots illustrate the RBP partners for XRCC6 (top), PPIL4 (middle), and LIN28B (bottom), with each point representing an individual RBP. The points are organized by the integrated distance from the specified RBP to the query RBP. Annotations within each plot designate the common function of the closest interacting partners. The three RBPs with the smallest distances are specifically labeled; those associated with a common function are marked in purple, and the others in gray. Source data are provided as a Source Data file. **C** Identification of RBP Partners of ZNF800 and TAF15: The swarm plots here delineate the RBP partners of ZNF800 (left) and TAF15 (right), employing the same color-coding for datasets as in (**A**): eCLIP (green), BioID (blue), and Perturb-seq (brown). The top portion represents the RBP partners as derived from individual datasets, each annotated with the common function of the nearest interacting partners. The bottom portion, analogous to (**B**), displays the RBP partners sorted by the integrated distance, with the top interacting RBPs distinctly labeled according

to the common function in purple and the others in gray. Source data are provided as a Source Data file. **D** Examination of RBP Partners of DGCR8: This section presents swarm plots of the RBP partners of DGCR8. The top plots showcase the partners based on individual source datasets, similar to (**C**), with each plot annotated and color-coded according to (**A**). The bottom plot displays the RBP partners sorted by integrated distance, highlighting the top interacting RBPs. Notably, TAF15 and XRN2 are emphasized, illustrating the efficacy of the distance integration procedure in confirming the known involvement of DGCR8 in the regulation of transcription. Source data are provided as a Source Data file. **E** Rearrangements in RBP matrices: This panel demonstrates the alterations in the structure of the Integrated Regulatory Interaction Map matrix due to random shuffling, depicting changes in distance to the closest and farthest partner RBP. Downsampling was conducted by shuffling distance values of varying fractions of RBPs (0% to 100%). This procedure was performed 10 times for each of 90 RBP, resulting in 900 estimates for each dataset and shuffling percent. Dots represent the median, error bars represent the lower and upper quartiles. Source data are provided as a Source Data file. **F** Percent of RBP pairs passing IRIM distance < 25% quantile that intersect STRING, OpenCell, hu.Map, and Zanzoni et al.[23]. Violin and boxplots are based on $10^4$ random shuffling iterations; red dots represent the percent of the real IRIM distances. Right-tailed $p$-values were obtained for each group by calculating a fraction of random shuffling iterations with the intersection greater or equal to the observed value (among $10^4 + 1$ cases). Box plot bounds and center represent the first, second, and third quartiles, while whiskers represent minimum and maximum values in the data, excluding outliers that are more than 1.5 interquartile range from lower and upper quartiles and are depicted as dots. Source data are provided as a Source Data file.

Indeed, XRCC6 participates in DNA repair pathways while also regulating rRNA biogenesis[28]. Interactions involving PPIL4 similarly point to its role in transcriptional regulation, a finding consistent with PPIL4 being shown to interact with JMJD6, a known actor in transcriptional control[29]. The interaction of LIN28B with other proteins also aligns with its recognized role in mRNA translation[30]. These examples highlight known interactions and hint at the potential for this label transfer approach to reveal previously unknown functions for RBPs, offering a starting point for further exploration.

As mentioned earlier, the incorporation of multiple data modalities allows IRIM to effectively capture the functional pleiotropy of RBPs. For illustration, we examined the annotated functions of the closest neighbors of each RBP across three modalities (Fig. 2C). Specifically, for ZNF800, its nearest neighbor in the eCLIP dataset–UCHL5–is identified as a chromatin remodeling protein. However, proximity labeling data reveals ribosome biogenesis factors like DDX21 and RPS11 in ZNF800's vicinity. Consequently, IRIM merges these modalities, revealing ZNF800's association with both chromatin remodeling and ribosome biogenesis factors (Fig. 2C, left panel). Another example is TAF15; eCLIP data link it to transcriptional regulators like SAFB2, while proximity labeling highlights its interaction with the splicing machinery through SMNDC1, GPKOW, and SF3B4. Perturb-seq data further captures translational regulators PABPC4 and EIF4G2 among TAF15's neighbors (Fig. 2C, right panel). Our method also discerns weak, yet consistent interactions between modalities. For instance, TAF15 and XRN2 show only distant connections with DGCR8 in individual modalities but are top-5 RBP partners of DGCR8 once the scores are integrated (Fig. 2D). Experimental evidence supports DGCR8's role in chromatin organization and its collaboration with XRN2 in transcription termination[31–33].

### Defining functional RBP neighborhoods using BioID-mediated proximity labeling

Having defined the modules that each RBP participates in, we next sought to assign regulatory functions to each of these modules. The proximity labeling data allowed us to go beyond RBP-RBP interactions (Supplementary Fig. S6A–D) and study the functions of both individual

RBPs and their modules by analyzing the totality of their protein neighborhoods (Supplementary Fig. S6F). For each RBP, we ranked its neighbors by their enrichment in the biotinylated fraction, followed by gene-set enrichment analysis (GSEA) to identify the most over-represented pathways and protein complexes in each RBP neighborhood (Fig. 3A). This procedure allowed us to systematically estimate the significance of the involvement of an RBP in a given pathway across all "RBP-pathway" pairs. Conceptually, the resulting GSEA $p$-values for the positive enrichments (enrichment scores > 0) reflect the confidence in each annotation, where higher -log($p$-values) denote higher confidence in the proposed association (Supplementary Fig. S6E, G, see "Methods"). We have visualized the high-confidence annotations in a heatmap (NES > 2 for at least one RBP) along with the major RNA classes that our eCLIP analysis nominated as the likely targets of each RBP module in Fig. 3B.

In many cases, the established functions of RBPs are clearly captured by this approach (Fig. 3B). For example, we have correctly annotated SRSF7, NONO, and HNRNPA1 as splicing-related RBPs that bind predominantly pre-mRNAs. Similarly, we identified RPS11, NPM1, and DDX52 as RBPs that are involved in ribosome biogenesis and directly interact with rRNAs. Our BioID-based annotations also identified RBPs that regulate transcription (HNRNPC, NPM1, QKI)[34–36], initiate and regulate mRNA translation (LARP4, EIF3G, RPS3, LIN28B)[30,37,38], participate in snRNA processing (TAF15, NPM1)[39,40] and mitochondrial metabolism (SUPV3L1, FASTKD2, TBRG4)[41], and modulate centrosome amplification (YWHAG)[42].

Our findings also reveal novel and previously unexplored "non-canonical" functions for human RBPs, highlighting the gaps in our current knowledge of RBP annotations that can be systematically addressed with our approach. For example, SRSF7 is primarily known as a splicing factor; however, we observed an equally strong enrichment of mRNA 3'-end processing and polyadenylation pathways, which are not yet annotated in GO but are alluded to in recent publications[43,44]. Overall, we have annotated 19 RBPs with 1111 BP GO terms at 5% FDR, of which 736 (66%) are novel (not listed in GO). In the following sections, we have experimentally verified a number of these annotations.

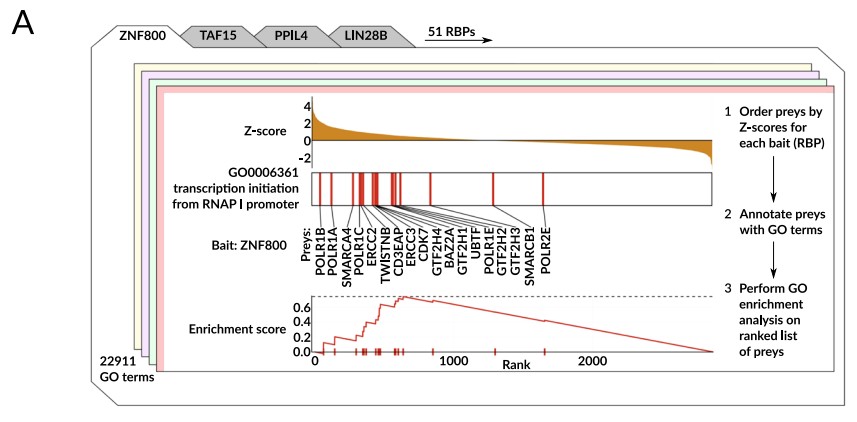

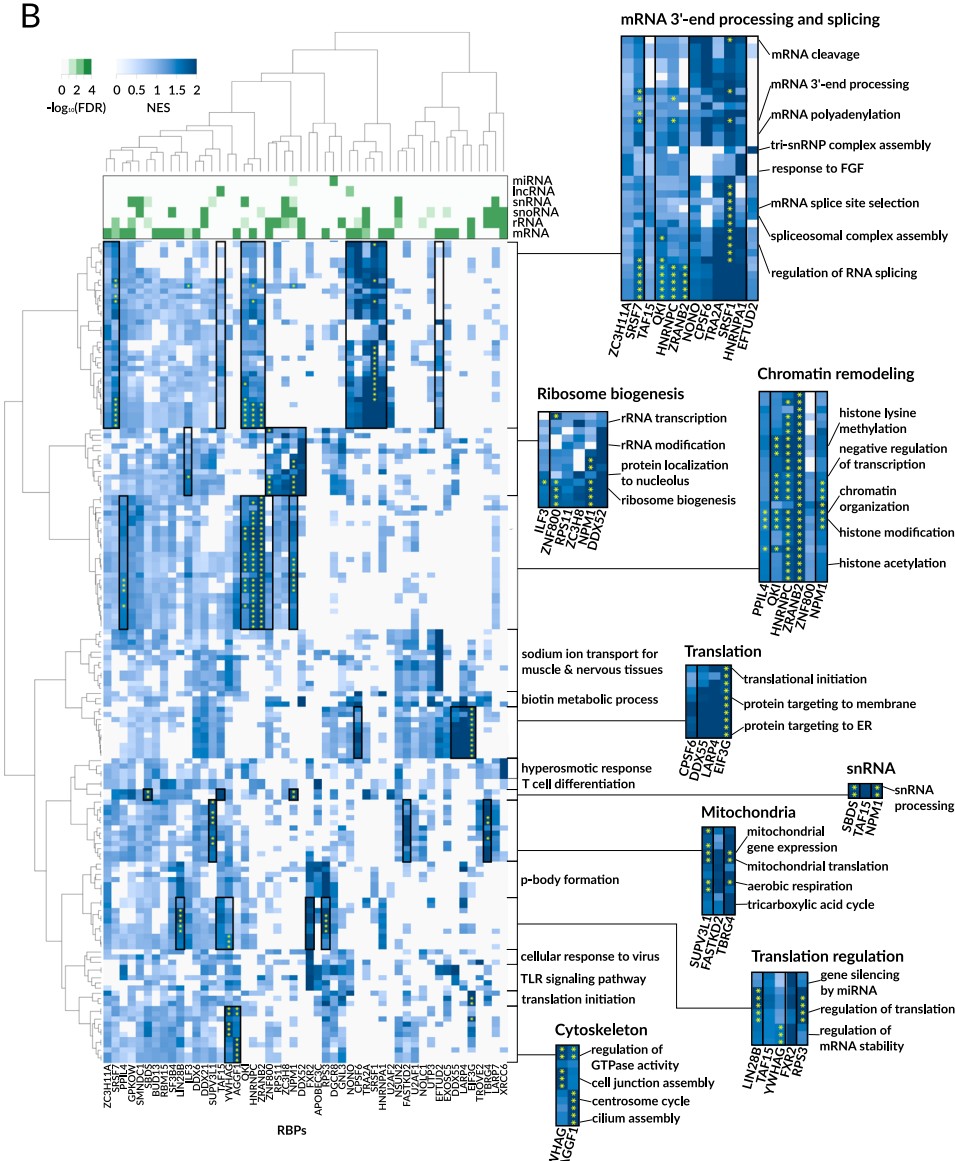

## ZC3H11A and TAF15 are pleiotropic post-transcriptional regulators involved in splicing, translation, and RNA stability

Our approach highlighted ZC3H11A and TAF15 as functionally pleiotropic RBPs involved in multiple post-transcriptional processes for distinct RNA regulons. ZC3H11A is known to be a part of the TREX complex responsible for mRNA export and was shown to co-localize with SRSF2 in nuclear speckles that play a role in splicing[45,46]. TAF15, on

the other hand, has been predominantly studied as part of the TFIID and RNAPII complexes of the core transcriptional machinery[47]. TAF15's role as an RNA-binding protein in post-transcriptional control, however, is not as well characterized. Regardless, several studies have shown TAF15 to be involved in the stability and processing of the mRNAs encoding FGFR4, GRIN1, and a small subset of other proteins in neurons, as well as lncRNA LINC00665[48–51]. In addition, it has been

**Fig. 3 | BioID-mediated proximity labeling defines RBP neighborhoods and enables functional annotation of RBPs. A** Overview of our pathway annotation workflow for RBPs. The example provided shows the test for the association of ZNF800 and GO:0006361 (transcription initiation from RNA polymerase I promoter). Proximity-labeled proteins were ranked by their z-scores in the ZNF800-BioID dataset, where a higher score implies enrichment relative to control. Experiments were performed in biological triplicates using unlabeled samples as controls (three cases vs. three control designs). Gene-set enrichment analyses were performed on the resulting ranked list across all RBPs. Each enrichment analysis

resulted in a p-value and NES score for a given pair of RBP and a pathway. **B** A heatmap showing the associations between RBPs and pathways as inferred from proximity labeling data. Columns correspond to the RBPs, rows correspond to individual gene ontology terms (Biological Processes; BP), and the color denotes the GSEA normalized enrichment score (NES). The associations showing FDR < 0.05 are marked with a yellow asterisk. The green heatmap in the header shows the RBP binding preferences to particular RNA types, as determined based on eCLIP RNA targets. Some known functions of RBPs are highlighted by boxes and zoomed-in on the right. Source data are provided as a Source Data file.

shown that TAF15 participates in miRNA-mediated regulation of cell cycle gene expression, and a role for this protein in mRNA transport and translation has been suggested based on its pervasive binding to 3'UTRs[51,52].

IRIM suggests that both ZC3H11A and TAF15 are involved in a much wider set of post-transcriptional regulatory processes than have been previously characterized (Fig. 2). In particular, TAF15 has the highest interaction scores with FUS, SAFB2, EIF4G2, NONO, and SAFB, which in addition to transcription are also associated with translation (EIF4G2) and splicing (NONO). On the other hand, ZC3H11A's top interacting partners include GPKOW and DHX30, suggesting putative splicing-related functions. Consistently, gene-set enrichment analysis of the proximity-labeling data revealed mRNA export (for ZC3H11A) and transcription (for TAF15), as the highest-scoring pathways. However, we also noted multiple additional high-scoring pathways, including "spliceosomal snRNP assembly" for both RBPs (GO:0000387, Normalized ES (NES) = 1.5) as well as "mRNA stabilization" (GO:0048255, NES = 1.5) and "positive regulation of translation" (GO:0045727, NES = 1.7) for TAF15 (Fig. 4A).

To verify these putative roles for ZC3H11A and TAF15 in splicing regulation, we used CRISPRi to knock down these RBPs in K562 cells (96% and 98% knockdown efficiency when compared to non-targeting guide RNA, respectively, Supplementary Fig. S7A–F) and performed paired-end total RNA-seq to evaluate transcriptome changes in response to RBP depletion. Upon silencing either of these genes, we observed a number of significant alternative splicing events (ASEs) (296 and 190 differentially spliced events for ZC3H11A and TAF15 knockdowns, respectively; Fig. 4B, D). We validated several of these significant ASEs (Supplementary Fig. S7G–I) using quantitative RT-PCR (Fig. 4C, E); thereby confirming the involvement of ZC3H11A and TAF15 in the regulation of alternative splicing. To confirm whether these modulations are the result of direct interaction between ZC3H11A or TAF15 and these target pre-mRNAs in vivo, we performed crosslinking and immunoprecipitation followed by sequencing (CLIP-seq)[53] for both ZC3H11A and TAF15 in K562 cells. As expected, we detected the binding of ZC3H11A at sites proximal to 326/353 ASEs (at the distance of <50 nt) and the binding of TAF15 at sites proximal to 202/218 ASEs (Fig. 4F and Supplementary Fig. S7J). Taken together, these results establish ZC3H11A and TAF15 as direct regulators of alternative splicing for their respective regulons.

In addition to RNA processing and splicing, we also observed a significant and independent association between TAF15 and translational control machinery. To investigate this, we performed ribosome footprinting (Ribo-seq)[54] as well as matched RNA sequencing in control and TAF15 knockdown cells (Supplementary Fig. S8A–C). Consistent with a direct role for TAF15 in translational control, we observed translational repression of 212 mRNAs in TAF15 knockdown cells (Supplementary Fig. S8C). Notably, these translationally repressed mRNAs were significantly enriched for RNAs that directly bind TAF15[18] (Fig. 5A). In addition, we generated and compared protein abundance data in TAF15 KD and control cell lines using quantitative mass spectrometry. As expected, TAF15 targets showed a significant change in their protein abundance without a concomitant change in their mRNA levels (Fig. 5A, B).

Taken together, our findings demonstrate that TAF15 plays a role in promoting mRNA translation for its target regulon.

We also observed a strong association between TAF15 with regulators of RNA stability including LARP1, SYNCRIP, and RBM10. To further explore this association, we measured mRNA decay rates by inhibiting RNAPII-mediated transcription with α-amanitin (Sigma-Aldrich A2263) and performing RNA-seq in control and TAF15 knockdown cells (Supplementary Fig. S8D–F)[55]. Using iPAGE[56] and GSEA[57], we found that TAF15-bound RNA targets are enriched among the RNAs that experience a reduction in half-life when TAF15 is depleted (Fig. 5C). To independently verify this observation, we used RT-qPCR to compare mRNA stability of several TAF15 mRNA targets, such as UBE2J2 and GUK1, in TAF15-KD versus control cells (Fig. 5C, D). For all tested targets, we observed significantly lower mRNA stability upon TAF15 knockdown, thus supporting our hypothesis of TAF15 involvement in the regulation of mRNA stability.

Collectively, these results showcase how a single RBP, in this case TAF15, can play multiple regulatory functions based on the context of its interactions with each regulon. To further explore this notion, we asked whether the three sets of TAF15 target RNAs, corresponding to TAF15's roles in splicing, translation, and stability, are in fact, distinct, form independent regulons, and participate in different biological processes (Fig. 5E and Supplementary Fig. S8G). We observed that the translation and stability regulons partially but significantly overlap; this is concordant with the known interdependence of these two biological processes[58]. On the other hand, there was only a small number of overlapping target RNAs present in the translation and splicing groups, as well as the splicing and stability groups (18 out of 741 and 40 out of 1890 genes, respectively; Fig. 5E). Overall, in K562 cells, TAF15 controls splicing of 155 RNAs, translation of 919 RNAs, and stability of 2068 RNAs; 320 of these RNAs fall into two regulons and only 13 are present in all three pathways, underscoring TAF15's involvement in three distinct regulatory pathways with largely mutually exclusive mRNA targets.

## RNA-binding proteins QKI and ZNF800 are involved in the regulation of transcription

While RNA-binding proteins are often thought to strictly regulate post-transcriptional processes, IRIM highlighted several RNA-binding proteins that are also strongly associated with transcriptional control. Chief among these, we noted ZNF800 and QKI; both associate with transcriptional regulators such as TAR DNA-binding Protein 43 (TDP-43)[59], Nucleophosmin 1 (NPM1)[60], and Helicase-Like Transcription Factor (HLTF). ZNF800 is a zinc finger protein whose molecular functions are poorly studied, yet it is implicated in diseases such as lung cancer[61]. In contrast, QKI is a well-studied RBP involved in many RNA-related processes and is known to play a major part in neuronal gene regulation and neuron myelination[62–66].

Based on our proximity labeling results, ZNF800's protein neighborhood functions in DNA methylation, transcription by RNA polymerase I, rRNA processing, and chromatin remodeling (Fig. 6A). On the other hand, QKI's neighborhood is associated with histone methylation, RNA splicing, transcription by RNA polymerase II, and chromatin organization. To validate the previously unknown role for

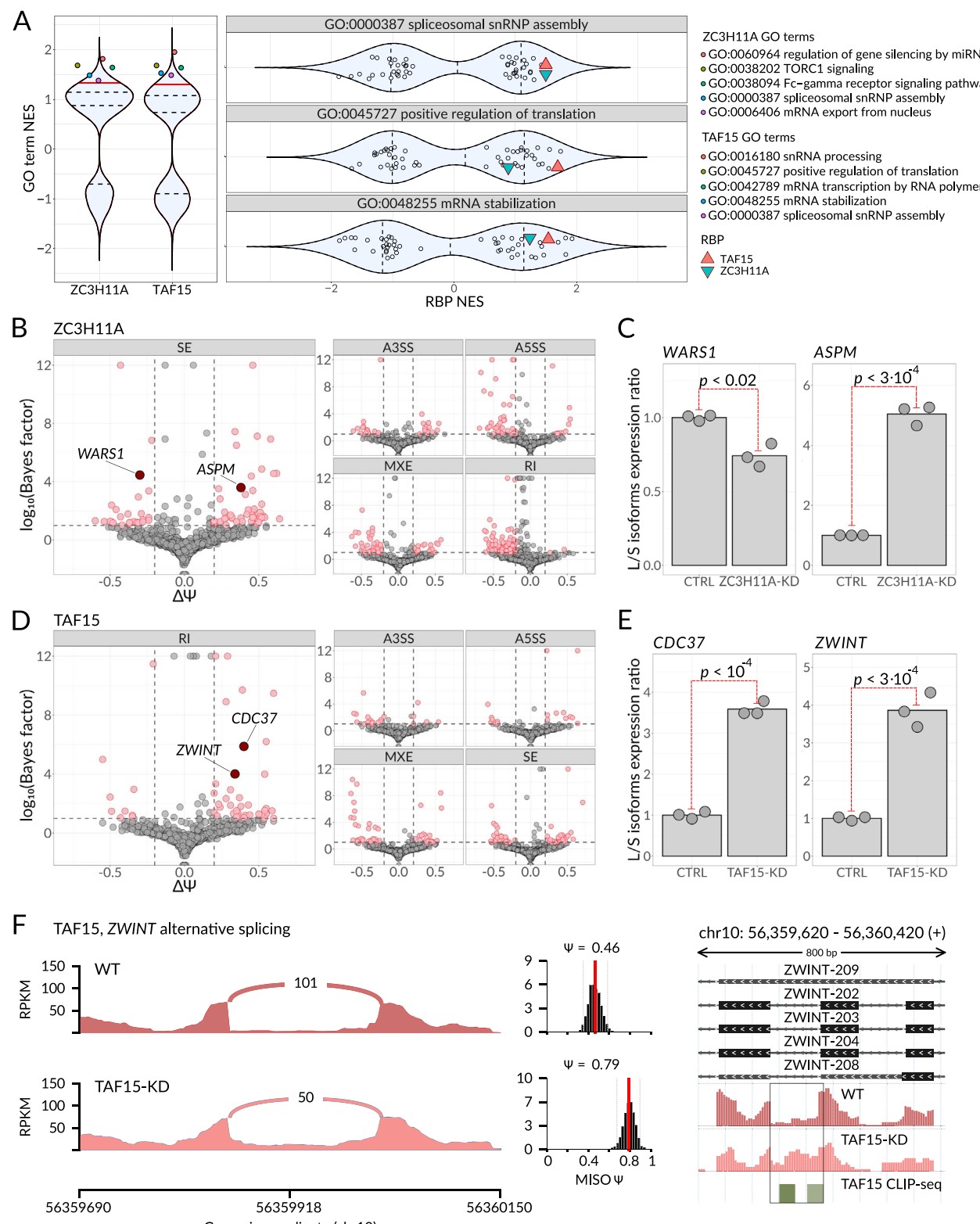

ZNF800 in chromatin remodeling and confirm recently discovered chromatin-associated QKI functions[36], we performed ATAC-seq on control and CRISPRi-generated knockdown K562 cells (Supplementary Fig. S9A–D) (87% and 76% knockdown efficiency, respectively)[67]. We observed a significant and widespread increase in chromatin accessibility across thousands of regions when these RBPs were silenced (2660 out of 2724 significantly differential regions were upregulated for QKI knockdown, and 1399 out of 1417 significantly differential

regions were upregulated in ZNF800 knockdown; Fig, 6B). Among the differentially accessible peaks, the majority were located in close proximity (< 1 Kb) to gene promoter regions (Fig. 6B).

To further demonstrate that ZNF800 and QKI are chromatin-associated RBPs, we performed ChIP-qPCR on several of their gene targets. Namely, we tested the binding of ZNF800 to the promoter sequences of *RPS15* and *RPL10A*, and the binding of QKI to the promoters of *PRC1* and *LTBR*. As expected, these target sequences were

**Fig. 4 | ZC3H11A and TAF15 control multiple independent regulons through distinct regulatory programs. A** Violin plots showing the normalized enrichment scores (NES) resulting from gene set enrichment analysis of proximity labeling data. Left subpanel: NES scores across all the GO-BP terms for ZC3H11A and TAF15 proteins. The five highest-scoring pathways are highlighted with color. Right subpanel: NES scores across all studied RBPs for the pathways GO:0000387, GO:0045727, and GO:0048255. ZC3H11A and TAF15 are highlighted with colored triangles. Dashed lines: quartiles; solid red line: 0.9 quantile. **B** Scatterplot showing changes in alternative splicing events (ASE) usage upon ZC3H11A knockdown as estimated by MISO. Individual subplots cover different classes of alternative splicing events: Skipped Exon (SE), Retained Intron (RI), Alternative 3′ Splice Site (A3SS), Alternative 5′ Splice Site (A5SS), and Mutually Exclusive Exon (MXE). Dashed lines indicate the following filters: Bayes factor ≥ 10 and the absolute value of isoforms levels difference ≥ 0.2. The ASEs passing these filters are shown in red. Source data are provided as a Source Data file. **C** Relative levels of two skipped exons from the

transcripts *WARS1* (left) and *ASPM* (right) were measured by RT-qPCR in control K562 and ZC3H11A-KD cells; $n = 3$ biological replicates. *P*-value from a one-sided *t* test performed on log-transformed isoform expression ratios, 0.0166 for *WARS1* and $2.86 \cdot 10^{-4}$ for *ASPM*. Source data are provided as a Source Data file. **D** Scatterplot showing changes in alternative splicing events in TAF15 knockdown cells, as in (**B**). Source data are provided as a Source Data file. **E** Relative levels of two retained introns from the transcripts *CDC37* (left) and *ZWINT* (right) were measured by RT-qPCR in control K562 and ZC3H11A-KD cells; $n = 3$ biological replicates. *P*-value from one-sided *t* test performed on log-transformed isoform expression ratios, $8.03 \cdot 10^{-5}$ for *CDC37* and $2.883 \cdot 10^{-4}$ for *ZWINT*. Source data are provided as a Source Data file. **F** Left: Sashimi plot illustrating the changes in intron retention event usage in *ZWINT* transcript upon TAF15 knockdown. Right: Genomic view of the *ZWINT* retained intron, RNA-seq profiles from WT and TAF15-KD cells, and TAF15 CLIP-seq peaks are shown at the bottom. *Y*-axis: counts per million (CPM). The region corresponding to the alternative splicing event is framed.

---

significantly enriched in ChIP samples compared to controls, which demonstrates the localization of ZNF800 and QKI to promoter regions of these identified target genes (Fig. 6C and Supplementary Fig. S9E, F). In addition to ChIP-qPCR validation, we have tested an overall agreement between the differential ATAC-seq peaks changing upon RBP knockdowns and the published ChIP-exo data[68]. As expected, ZNF800 and QKI ChIP-exo signal was significantly enriched in differential ATAC-seq peaks compared to the rest of the peaks (U test *p*-value $< 10^{-16}$ for ZNF800, Fisher's exact test odds ratio = 40, *p*-value $< 10^{-16}$ for QKI, see "Methods").

To test whether the observed changes in chromatin accessibility lead to changes in mRNA expression, we next performed RNA-seq in control and QKI- or ZNF800- knockdown cells (Supplementary Fig. S9A, C). As expected, we observed significantly elevated expression of the genes with increased chromatin accessibility in the ATAC-seq data (18X and 4X increase in median RNA-seq LogFC for ZNF800 and QKI ATAC-seq targets, respectively, Fig. 6D). Together, these observations point to the role of ZNF800 and QKI as transcriptional repressors.

We also sought to explore whether the role that ZNF800 and QKI play in transcription inhibition is associated with their binding to RNA. We tested whether the promoters of genes encoding the RNA binding targets of ZNF800 and QKI (based on eCLIP data) overlap the ATAC-seq peaks that become upregulated upon RBP knockdown. We observed that such overlapped ATAC-seq peaks were significantly more upregulated than the rest of the peaks (69% and 47% increase in median ATAC-seq LogFC for ZNF800 and QKI eCLIP targets, respectively, Fig. 6E), supporting the hypothesis that ZNF800 and QKI achieve their regulatory functions through direct co-transcriptional binding of chromatin-associated RNA.

Collectively, these results validate a direct and previously unknown role for QKI and ZNF800 in transcriptional control, as revealed by IRIM, even though they were previously thought to be primarily involved in post-transcriptional regulation. Our data suggest that RBP-RNA interactions can often influence transcriptional activity. This further highlights the value of the IRIM in identifying underappreciated functions of multimodal RBPs.

## Discussion

The traditional model of transcriptional control called the "transcriptional regulatory code"[69] involves *cis*-acting elements such as enhancers and transcription factor binding sites (TFBSs) and *trans*-acting transcription factors (TFs) that bind to these elements in a combinatorial and coordinated manner to create complex regulatory circuits. However, the equivalent conceptual framework for studying the combinatorial post-transcriptional control of gene expression has not been established. Given that a few hundred RBPs control all aspects of the RNA life cycle, from processing and export to translation and

decay, the "one RBP-one function" paradigm does not provide enough complexity to cover all the post-transcriptional regulatory processes that occur in a cell. It is not surprising, then, that RBPs are highly functionally pleiotropic and also exhibit a complex and context-specific RNA binding grammar.

Many research initiatives have focused on mapping RBP-bound transcripts as units of post-transcriptional gene expression control. The ENCODE consortium and other groups have used methods like eCLIP and RIP-seq for this purpose[7,70]. While these efforts have provided valuable insights, they often do not capture the full complexity of RBP functions, which are multifaceted and context-dependent[71,72]. It's widely recognized that RBPs often bind thousands of RNAs, exhibiting regulatory functions that vary across different contexts. As such, considering an RBP regulon as a simple set of RNAs bound by a given protein is an oversimplification.

In this study, we took a significant step forward by providing detailed annotations of these combinatorial interactions. We refined the concept of 'regulatory modules,' not as a novel idea, but as a framework to systematize the complex interplay of RBPs. Our definition of regulatory modules—collections of RNA-binding proteins that work together for a specific function and regulate distinct sets of target RNAs—facilitates a deeper understanding of the many-to-many relationships between RBPs and their functions. This approach has enabled us to perform comprehensive mapping of RBP-RBP functional interactions, leading to the annotation of regulatory modules. Equally important, we have applied these annotations to deconvolve the totality of RBP-RNA binding events, often collated into a plan set of mRNA targets[18], into distinct regulons. This methodology not only enriched our understanding of RBP regulatory networks but also added specificity to existing datasets, offering a more nuanced view of post-transcriptional control mechanisms.

To further aid researchers in exploring our data, we have developed a Shiny app, [RBP Browser] (https://goodarzilab.shinyapps.io/RBP-Browser/), which offers an interactive map of human RNA-binding protein interactions. This tool allows users to query their RBP of interest and understand how it fits into the functional network of RNA regulation in human cells.

Instead of viewing post-transcriptional regulation through the lens of individual RBPs and their bound target RNAs, we propose that the field should instead adopt a more precise definition of RBP regulons that accounts for their context-specificity. To address this issue, we propose the concept of "regulatory modules" as the foundational units of post-transcriptional control, i.e., collections of RNA-binding proteins that work together for a specific function and a distinct target regulon. This approach allows us to capture the many-to-many relationship between RNA-binding proteins and their regulatory functions. In this work, we performed large-scale mapping of RBP-RBP functional interactions, which then allowed us to map the regulatory modules,

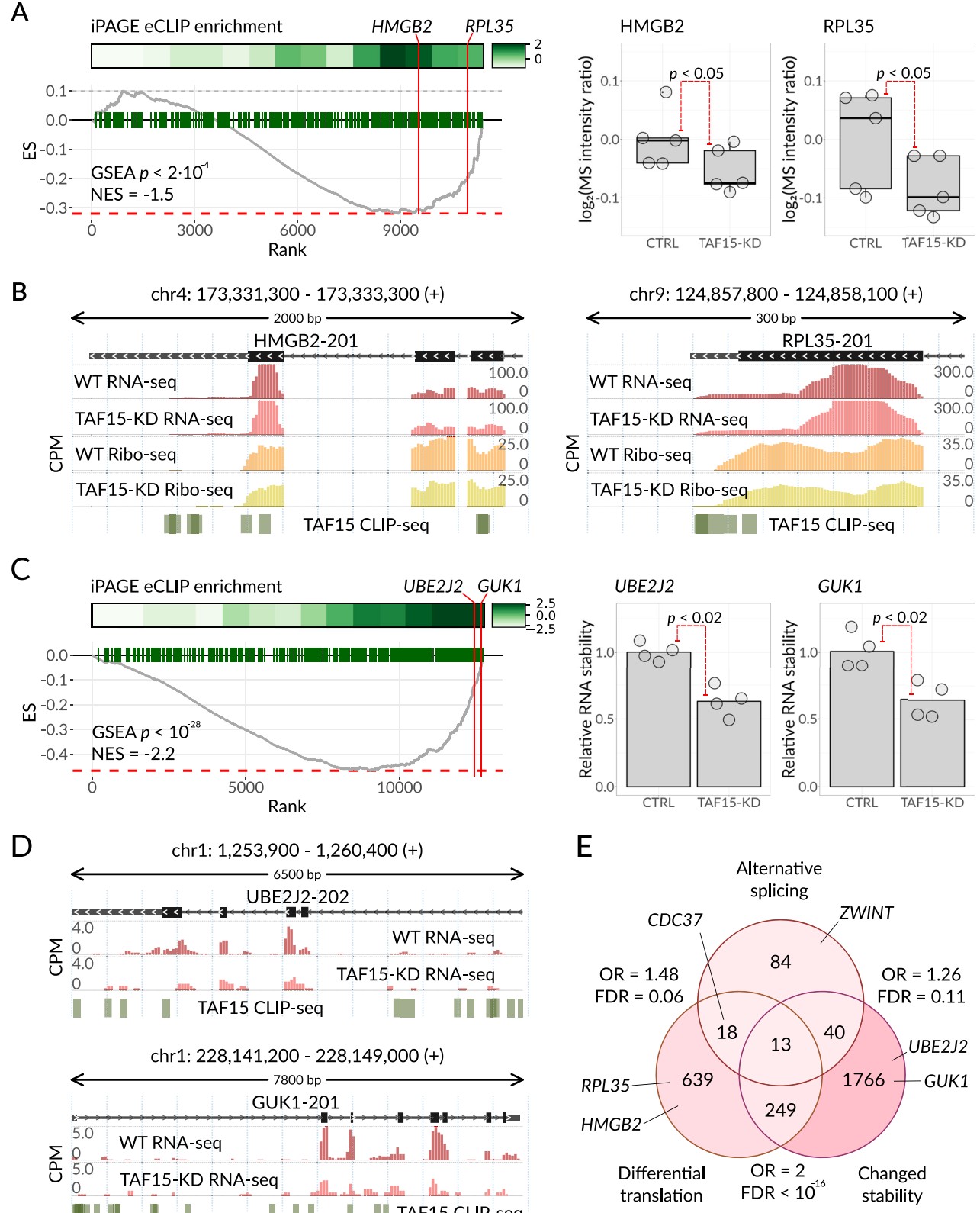

and annotate their associated functions. Through this annotation process, we discovered that multiple proteins govern independent regulons, each with distinct functions. Among these, TAF15, ZC3H11A, ZNF800, and QKI were biochemically validated to demonstrate their roles in governing such regulons. This aspect of our study emphasizes the functional pleiotropy of individual RBPs and significantly broadens our understanding of their diverse regulatory roles.

In this study, our use of BioID labeling-based pulldown followed by mass spectrometry has been instrumental in mapping the protein neighborhoods of RBPs, providing a deeper insight into their interactions within cellular networks[17]. The key advancement in our methodology was the inclusion of matched pulldown controls for each of the 50 human RBPs analyzed. This methodological precision, not commonly found in similar studies, significantly improved the reliability of

**Fig. 5 | TAF15 is directly involved in RNA translation and stability regulation.**
**A** Left: enrichment analysis of TAF15 mRNA targets among the differentially translated genes (in the TAF15-KD cell line compared to the WT cell line). The differential ribosome occupancy (RO) measurements in TAF15-KD cells were estimated from Ribo-seq. The genes were sorted based on the RO change (along the *x*-axis), and the enrichment of TAF15 mRNA targets, inferred from eCLIP data, was calculated using iPAGE (top subpanel) and with GSEA (bottom subpanel, ES stands for the enrichment score). Two example targets, HMGB2 and RPL35, are highlighted. Right: levels of HMGB2 and RPL35 were measured by mass spectrometry in control K562 and TAF15-KD cells. $N = 5$ biological replicates. *P*-value from one-sided Wilcoxon rank sum test, 0.04762 for both HMGB2 and RPL35. Source data are provided as a Source Data file. **B** Genomic view of *HMGB2* (left) and *RPL35* (right). RNA-seq and Ribo-seq WT and TAF15-KD profiles, as well as TAF15 CLIP-seq peaks, are shown below. *Y*-axis: counts per million (CPM). **C** Left: enrichment analysis of TAF15 mRNA targets among the differentially stabilized transcripts (in TAF15-KD cell line compared to WT cell line) measured by α-amanitin treatment. The

transcripts were sorted based on stability change (log2FCs). The enrichment of TAF15 RNA targets, inferred from eCLIP data, was calculated with iPAGE (top and middle subpanel) and with GSEA (bottom subpanel). Two example targets, UBE2J2 and GUK1, are highlighted. Right: relative stability of *UBE2J2* and *GUK1* mRNAs were measured as mRNA to pre-mRNA abundances ratio using qPCR in control K562 and TAF15-KD cells. $N = 4$ biological replicates. *P*-value from one-sided Wilcoxon rank sum test, 0.01429 for *UBE2J2* and 0.0147 for *GUK1*. Source data are provided as a Source Data file. **D** Genomic view of *UBE2J2* (top) and *GUK1* (bottom). RNA-seq WT and TAF15-KD profiles, as well as TAF15 CLIP-seq peaks, are shown below. *Y*-axis: counts per million (CPM). **E** Venn diagram of TAF15 RNA regulons. Shown are the numbers of genes that exhibit significant changes in splicing (155 genes with Bayes factor ≥ 10), translation (919 genes with FDR < 0.05), or stability (2068 genes with FDR < 0.05) upon TAF15 knockdown, as captured by RNA-seq, Ribo-seq, and RNA-seq with α-amanitin, respectively. Results of one-sided Fisher's exact test for each pairwise intersection were FDR-corrected for multiple testing and are shown next to the corresponding area. Source data are provided as a Source Data file.

our data[10,73,74]. The reliability of our pulldown profiles, evidenced by their closer resemblance to matched negative controls than to other pulldown profiles (Supplementary Fig. S6D), was essential for accurately dissecting the complex interplay of these proteins. This accuracy is crucial, particularly as high-throughput methodologies often lead to false annotations[75]. The methodological rigor becomes even more critical as the field moves towards understanding the dynamic nature of RBP functions and interactions. Looking ahead, integrating technologies like live-cell imaging or time-resolved mass spectrometry could further enrich our understanding, adding a temporal dimension to RBP regulatory networks. Our rigorous approach lays the groundwork for future dynamic and integrated studies of gene expression regulation, offering valuable insights and testable hypotheses to the scientific community.

The dynamic nature of functions and interactions within cellular networks, continually evolving in response to various cellular conditions and stimuli, is increasingly recognized as a critical aspect of genomics research[16]. Our study's emphasis on multi-omics integration aligns with this evolving paradigm, enabling us to capture a wide spectrum of interactions within the complex RBP regulatory networks. While our current approach offers a comprehensive snapshot, the next frontier in the field involves delving deeper into the dynamic behaviors of these interactions. Future research should focus on integrating methodologies that can track these changes over time, providing insights into how these interactions fluctuate and respond to different cellular stimuli. Such advancements will be instrumental in fully deciphering the nuanced and ever-changing landscape of RBP-mediated gene regulation.

The current study encompasses approximately ~100 out of the estimated ~1000 RNA-binding proteins. While we have carefully selected RBPs to cover a variety of functional pathways and subcellular compartments, expanding this dataset to include a broader spectrum of RBPs is essential for a more holistic understanding of the regulatory network.

A key limitation of our proximity labeling methodology is the transgene expression of fusion proteins, which could potentially alter protein expression, localization, and, consequently, their function. The overexpression of fusion proteins could potentially lead to artifacts in protein interaction data, although our spot checks did not reveal significant localization changes. The immunofluorescence assays performed on a subset of five RBPs suggested that transgene expression did not markedly affect the native behavior of these proteins. Moreover, the RBP interaction data was consistent with the OpenCell dataset[5], which was generated using endogenous tagging. Broadly, the RBP overexpression, the fusion of BioID2 protein to the RBP, and the focus on a single isoform per protein may introduce biases in the collected data.

A limitation of the Perturb-seq assay is the limited cell sampling per perturbation, which might impact the breadth of data representation. Despite this, our statistical analyses have shown that the data is robust even with added noise. Expanding the cell numbers or sequencing depth in future studies would not only confirm these findings but also enhance the statistical power and comprehensive representation of these analyses.

Lastly, the data generated through our high-throughput approach primarily serves as a foundation for hypothesis generation. While the IRIM and the mapped regulons provide valuable insights, they represent a starting point for detailed mechanistic studies. Future research should focus on experimentally validating and extending these findings to unravel the complex dynamics of RBP-mediated regulation.

IRIM underscores that functionally, pleiotropic RBPs playing different and even divergent roles depending on their specific context are the rule, rather than the exception. The deconvolved RNA regulons provide a set of readily testable hypotheses for the scientific community. In addition, the datasets generated in this study serve as a valuable resource for further exploration of RBP functions. Studying the role of RBPs in gene expression regulation necessitates a deeper understanding of complex combinatorial interactions between these proteins. This study represents a significant step towards building a comprehensive and integrated framework for examining these intricate regulatory mechanisms.

## Methods

### Cell lines

All cells were cultured in a 37 °C 5% CO2 humidified incubator. The 293 T cells (ATCC CRL-3216) were cultured in DMEM high-glucose medium supplemented with 10% FBS, glucose (4.5 g/L), L-glutamine (4 mM), sodium pyruvate (1 mM), penicillin (100 units/mL), streptomycin (100 μg/mL) and amphotericin B (1 μg/mL) (Life Technologies Corporation 15290026). The K562 cell line (ATCC CCL-243) was cultured in RPMI-1640 medium supplemented with 10% FBS, glucose (2 g/L), L-glutamine (2 mM), 25 mM HEPES, penicillin (100 units/mL), streptomycin (100 μg/mL) and amphotericin B (1 μg/mL) (Life Technologies Corporation 15290026). All cell lines were routinely screened for mycoplasma with a PCR-based assay.

### BioID2-RBP fusion cell line generation

50 RBPs were selected based on the 3 criteria: (i) ENCODE eCLIP data availability for a given RBP, (ii) presence of a given RBP in the ORFeome entry clone library[76], (iii) representing diverse RNA metabolic processes. In order to construct the cell lines stably expressing BioID2-RBP fusion proteins, we first cloned in an open reading frame of BioID2 enzyme[17], followed by a linker (YPAFLYKVVYGGGGSGGGGSGGGGS) and attR-flanked *ccdB* counterselection marker for Gateway cloning,

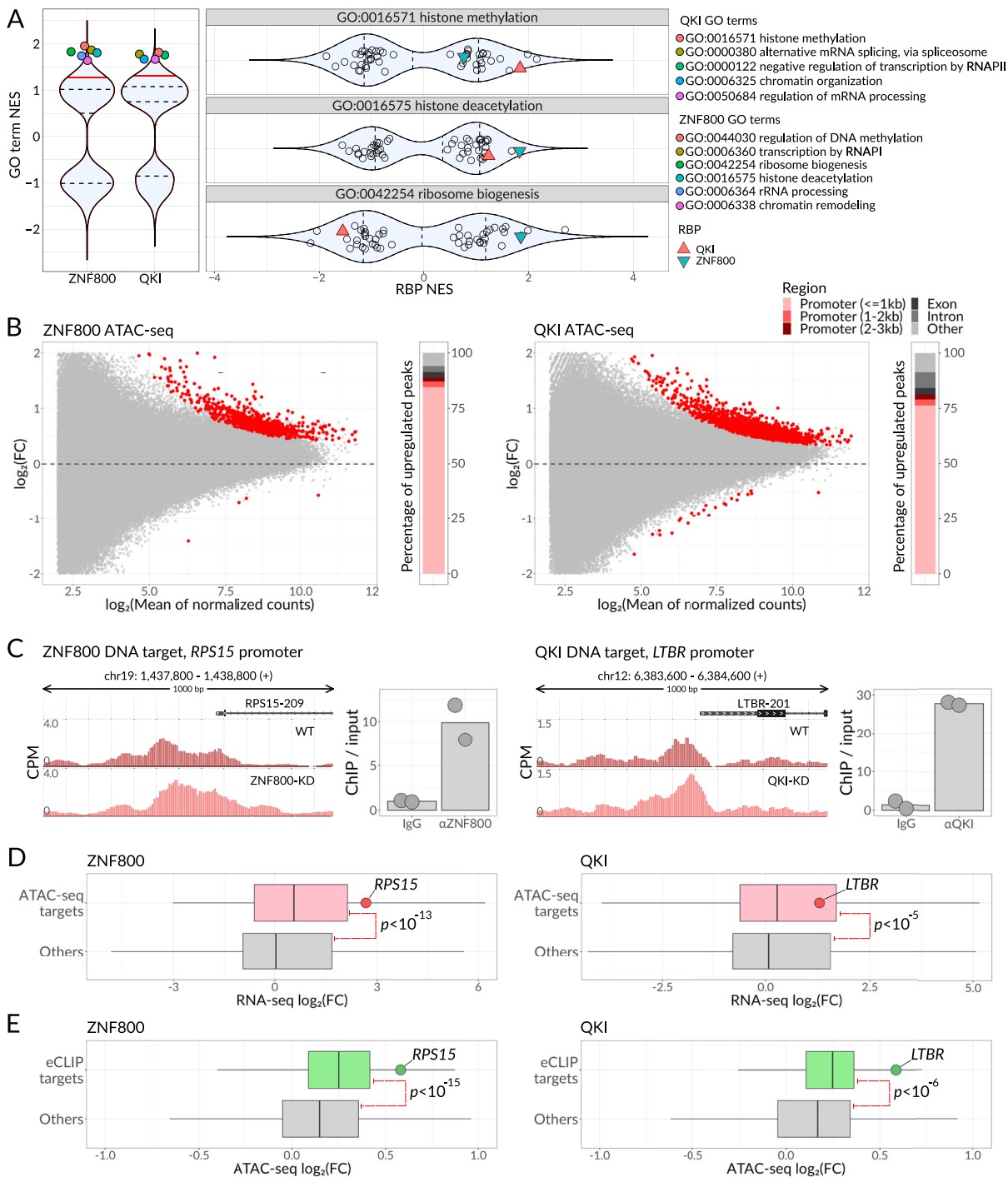

into the pWPI backbone (Addgene #12254). The resulting backbone is named pWPI_GW_BioID2_T2A_Blast (Addgene #135448) and is available on Addgene (#214831). We then used Gateway LR Clonase II Enzyme mix (Thermo Fisher 11791020) to clone the open reading frames of the RBPs of interest (from ORFeome entry clone library[76]) into the destination vector. The lentiviral constructs were co-transfected with pCMV-dR8.91 and pMD2.D plasmids using NanoFect (ALSTEM NF100) into 293 T cells (ATCC CRL-3216), following the manufacturer's protocol. The virus was harvested 48 hours post-transfection and passed through a 0.45 μm filter. K562 cells (ATCC CCL-243) were then transduced for 2 h while centrifuging (800 RPM) with the filtered virus in the presence of 8 μg/mL polybrene (Millipore C788D57). Cells were selected with 20 μg/mL blasticidin (Gibco A1113903) for 5 days. The expression of the fusion protein was validated by western blotting.

## Western blotting

Cell lysates were prepared by lysing cells in ice-cold RIPA buffer (25 mM Tris-HCl pH 7.6, 0.15 M NaCl, 1% IGEPAL CA-630 (Sigma-Aldrich 9002-93-1), 1% sodium deoxycholate, 0.1% SDS) (Sigma-Aldrich SIAL-R0278-50ML) containing 1X protease inhibitors (Thermo Fisher Scientific PI78410). Lysate was cleared by centrifugation at $20{,}000 \times g$ for 10 min at 4 °C. Samples were denatured for 10 min at 70 °C in 1X LDS loading buffer (Invitrogen/Fisher Scientific NP0007) and 50 mM DTT (Scientific Laboratory Supplies Ltd NAT1068). Proteins were separated by SDS-PAGE (Invitrogen/Fisher Scientific P2325) using 4–12% Bis-Tris NuPAGE gels (Thermo Fisher Scientific NP0321BOX), transferred to nitrocellulose (Millipore WP2HY315F5), blocked using 5% BSA (VWR International 97064-340), and probed using target-specific antibodies. Bound antibodies were detected using dye-conjugated secondary

**Fig. 6 | ZNF800 and QKI control gene expression at transcriptional and post-transcriptional levels independently. A** Violin plot showing the normalized enrichment scores (NES) resulting from gene set enrichment analysis of proximity labeling data. Left subpanel: NES scores across all the GO-BP terms for ZNF800 and QKI proteins. The 5 highest scoring pathways are highlighted with color. Right subpanel: NES scores across all the studied RBPs for GO:0016571, GO:0016575, and GO:0042254 GO terms. ZNF800 and QKI are highlighted with colored triangles. Dashed lines: quartiles; solid red line: 0.9 quantile. **B** Volcano plots showing differential chromatin accessibility between WT K562 cells and ZNF800-KD (left) or QKI-KD (right) cells. Each point denotes a single ATAC-seq peak; peaks passing 0.1 FDR are colored red. The distribution of peaks among various genomic regions is shown on the right of each volcano plot. Source data are provided as a Source Data file. **C** Genomic view of *RPS15* (left) and *LTBR* (right) promoter regions. ATAC-seq profiles of WT cells along with ZNF800-KD (left) or QKI-KD (right) are shown. The Binding of ZNF800 to the *RPS15* promoter region and the binding of QKI to the *LTBR* promoter region were measured by ChIP-qPCR in K562 cells and are illustrated on the right of each profile plot. Source data are provided as a Source Data file. **D** Box plots showing the distributions of expression fold changes in WT cells compared to either ZNF800-KD cells (left) or QKI-KD cells (right), as measured by RNA-seq. The distributions for the genes showing significant promoter accessibility increase upon the respective knockdown and for the rest of the genes are shown separately. The top most highly accessible ATAC-seq peak was considered for each gene resulting in 21708 genes in both ZNF800-KD and QKI-KD cells, of which 834 (3.8%) and 1476 (6.8%) had their promoters accessibility increased upon ZNF800 and QKI knockdown, respectively. *P*-value calculated by one-sided Wilcoxon rank sum test, $8.1 \cdot 10^{-14}$ for ZNF800-KD and $2.64 \cdot 10^{-6}$ for QKI-KD. Box plot bounds and center represent the first, second, and third quartiles, while whiskers represent minimum and maximum values in the data, excluding outliers that are more than 1.5 interquartile range from lower and upper quartiles and are depicted as dots. Source data are provided as a Source Data file. **E** Box plots depicted as in (**D**) showing the distributions of chromatin accessibility fold changes in WT cells compared to either ZNF800-KD cells (left) or QKI-KD cells (right), as measured by ATAC-seq. The distributions for ZNF800- or QKI- RNA targets (as defined by eCLIP) and the rest of the genes are shown separately. In total, there are 23275 ATAC-seq peaks, with 714 assigned to ZNF800 RNA target genes (leaving 22561 as non-target) and 286 assigned to QKI RNA target genes (leaving 22989 as non-target). *P*-value calculated by one-sided Wilcoxon rank sum test, $6.81 \cdot 10^{-20}$ for ZNF800-KD and $2.3 \cdot 10^{-7}$ for QKI-KD. Source data are provided as a Source Data file.

antibodies according to the manufacturer's instructions. Antibodies: HA (BioLegend 901533), eIF3I (BioLegend 646701), beta-tubulin (Proteintech 10094-1-AP), GAPDH (Proteintech 10494-1-AP). The uncropped images of western blots are provided in the Source Data File.

### Biotin treatment and pulldown

The pulldown was performed as described in ref. [17]. Cells were incubated with biotin-depleted media (biotin-free RPMI-1640 medium, supplemented with 10% dialyzed FBS, glucose (2 g/L), L-glutamine (2 mM), 25 mM HEPES, penicillin (100 units/mL), streptomycin (100 μg/mL) and amphotericin B (1 μg/mL) for 72 h before analysis. For BioID2 pulldown, $12 \times 10^6$ cells per replicate were incubated with 50 μM biotin for 24 h. For the negative control samples, $12 \times 10^6$ cells per replicate were incubated with DMSO. After three times of PBS washes, the cells were lysed in 1 ml of lysis buffer containing 50 mM Tris, pH 7.5, 150 mM NaCl, 1 mM EDTA, 1 mM EGTA, 1% Triton X-100, 1% Sodium deoxycholate, 0.1% SDS, 1 × Complete protease inhibitor (Halt Phosphatase Inhibitor Cocktail; Thermo Fisher Scientific 78420), and 250 units benzonase (EMD Millipore 706643). The lysates were passed through a 25 G needle 10 times and cleared 10 min at $14,000 \times g$ at $+4\,^\circ$C. The protein concentration was measured with BCA Protein Assay Kit (Thermo Scientific A55865); the lysate was diluted to a concentration of 2 μg/mL. 500 μl of lysate was incubated with 125 μl of Dynabeads (MyOne Streptavidin C1; Thermo Fisher Scientific 65001) overnight with rotation at $+4\,^\circ$C. Beads were collected using a magnetic stand and washed twice with 2% (wt/vol) SDS, twice with wash buffer containing 50 mM Tris, pH 7.5, 500 mM NaCl, 1 mM EDTA, 1 mM EGTA, 1% Triton X-100, 0.1% SDS, twice with wash buffer containing 50 mM Tris, pH 7.5, 150 mM NaCl, 1 mM EDTA, 1 mM EGTA, 1% Triton X-100, 0.1% SDS, then boiled for 5 min in 50 μl of elution buffer containing 2% SDS, 100 mM DTT (Scientific Laboratory Supplies Ltd NAT1068), Tris-HCl pH 7.5. The supernatant was collected and saved for mass spectrometry analysis.

### Mass spectrometry analysis

Eluted BioID samples were reduced by the addition of 100 mM DTT (Scientific Laboratory Supplies Ltd NAT1068) and boiling at 95 °C for 10 min before being subjected to Filter Aided Sample Preparation (FASP) to generate tryptic peptides, as described previously (Dermit et al. Dev Cell, 2020). Briefly, samples were diluted 7-fold in UA buffer (8 M urea, 100 mM Tris HCl pH 8.5) (Sigma-Aldrich U1250-5KG), transferred to Vivacon 500 Hydrosart centrifugal filters with a molecular cut-off of 30 kDa (Sartorius), and concentrated by centrifugation at $14,000 \times g$ for 15 min. Filters were then washed twice by the addition of 0.2 mL of UA buffer (Sigma-Aldrich U1250-5KG) to the filter tops and re-concentrating. Reduced cysteine residues were then alkylated by addition of 100 μL of

50 mM iodoacetamide (VWR International Ltd 786-228) dissolved in UA buffer (Sigma-Aldrich U1250-5KG), and incubation at room temperature in the dark for 30 min. The iodoacetamide solution was then removed by centrifugation at $14,000 \times g$ for 10 min, and samples were washed twice with 0.2 mL of UA buffer (Sigma-Aldrich U1250-5KG)as before. Urea was then removed from samples by performing three washes with 0.2 mL of ABC buffer (0.04 M ammonium bicarbonate) (Sigma-Aldrich A64141-500G). Filters were then transferred to fresh collection tubes, and proteins were digested by the addition of 0.3 μg of MS grade Trypsin (Sigma-Aldrich T6567-1MG) dissolved in 50 μL of ABC buffer (Sigma-Aldrich A64141-500G), and overnight incubation in a thermo-mixer at 37 °C with gentle shaking (600 rpm). The resulting peptides were eluted from the filters by centrifugation at $14,000 \times g$ for 10 min. Residual remaining peptides were further eluted by the addition of 100 μL ABC (Sigma-Aldrich A64141-500G) to the filter tops and centrifugation. This was repeated once and the combined eluates were then dried by vacuum centrifugation (no heating) and reconstitution in 2% Acetonitrile (ACN) (VWR International Ltd 9012.1000GL), 0.2% Trifluoroacetic acid (TFA) (Life Technologies Ltd Invitrogen Division 85183), followed by desalting using C18 StageTips (Rappsilber, et al., Nat Protoc. 2007). The desalted peptides were dried again by vacuum centrifugation (45 °C) and re-suspended in A* buffer (2%ACN, 0.5% Acetic acid (Fisher Scientific UK Ltd 10171460), 0.1% TFA in water) before LC-MS/MS analysis. 1/3rd of each sample was analyzed on a Q-Exactive Plus Orbitrap mass spectrometer coupled with a nanoflow ultimate 3000 RSL nano HPLC platform (Thermo Fisher). Samples were resolved at a flow rate of 250 nL/min on an Easy-Spray 50 cm × 75 μm RSLC C18 column with 2 μm particle size (Thermo Fisher), using a 123 min gradient of 3% to 35% of buffer-B (0.1% formic acid in ACN) against buffer-A (0.1% formic acid in water), and the separated peptides were infused into the mass spectrometer by electrospray (1.95 kV spray voltage, 255 °C capillary temperature). The mass spectrometer was operated in data-dependent positive mode, with 1 MS scan followed by 15 MS/MS scans (top 15 method). The scans were acquired in the mass analyzer at 375–1500 m/z range, with a resolution of 70,000 for the MS and 17,500 for the MS/MS scans. A 30-s dynamic exclusion of fragmented peaks was applied to limit repeated fragmentation of the same ions.

### Perturb-seq

68 RBPs were chosen for Perturb-seq analysis based on the clustering analysis of the ENCODE eCLIP dataset and DeepBind dataset[77]. Perturb-seq experiment was performed as previously described[78]. Briefly, a library of 205 sgRNAs (5 non-targeting sgRNAs and 200 sgRNAs targeting 100 genes, 2 sgRNAs per gene) was ordered as a pooled oligonucleotide library from Twist Bioscience with the following design:

[ATCTTGTGGAAAGGACGAAACACCG]-[Protospacer Sequence]-[GTTTTAGAGCTAGAAATAGCAAGTTAAAATAAGGC]

The library was PCR-amplified using Q5 Hot Start High-Fidelity 2X Master Mix (NEB VWR International: 102500-140) with the primers with the following sequences: 5'-ATCTTGTGGAAAGGAC-3' and 5'-GCCTTATTTTAACTTGCTA-3'. To clone libraries into CROPseq-Guide-Puro vector (Addgene #86708), the starting vector was digested with BsmBI (Fisher Scientific FERER0451) following the protocol outlined in ref. 79. The library was cloned into the digested backbone using the Gibson Assembly method[80]. The reaction product was transformed into Takara Stellar competent cells according to manufacturer recommendations, grown overnight in 100 mL LB with ampicillin, and purified using ZymoPURE II Plasmid Midiprep Kit (Zymo Research D4200). K562 cells (ATCC CCL-243) were infected with the plasmid library at a low multiplicity of infection to minimize double infection. The infected cells were selected with 2 μg/mL puromycin (Gibco A1113802) for 3 days. Live cells were isolated on a flow cytometer (FACSAria II) by propidium iodide staining (Thermo Fisher Scientific P1304MP). Approximately 5000 live cells were captured by 10X Chromium Controller using Chromium Single Cell 3' Reagent Kits v2. Sample preparation was performed according to the manufacturer's protocol. Samples were sequenced on a NovaSeq 6000 using the following configuration: Read 1: 28, i7 index: 8, i5 index: 0, Read 2: 98.

To facilitate sgRNA assignment, sgRNA-containing transcripts were additionally amplified by PCR reactions by modifying a previously published approach[81]. The following primers were used for amplification: 5'-AATGATACGGCGACCACCGAGATCTACAC-3' and 5'-CAAGCAGAAGACGGCATACGAGATTACGACAGGTGACTGGAGTTCA-GACGTGTGCTCTTCCGATCTggactatcatatgcttaccgtaacttgaaag-3'. PCR product was cleaned up by 1.0x SPRI beads (SPRIselect; Beckman Colter B23317). Samples were sequenced using paired-end 150 bp sequencing on an Illumina MiSeq sequencer.

## CRISPRi-mediated gene knockdown
K562 cells (ATCC CCL-243) expressing dCas9-KRAB fusion protein were constructed by lentiviral delivery of pMH0006 (Addgene #135448) and FACS isolation of BFP-positive cells.

The lentiviral constructs were co-transfected with pCMV-dR8.91 (Andwin Scientific NC2092494) and pMD2.D (Addgene #12259) plasmids using TransIT-Lenti (Mirus 75814-982) into 293 T cells (ATCC CCL-3216), following the manufacturer's protocol. The virus was harvested 48 hours post-transfection and passed through a 0.45 μm filter. Target cells were then transduced overnight with the filtered virus in the presence of 8 μg/mL polybrene (Millipore C788D57).

Guide RNA sequences for CRISPRi-mediated gene knockdown were cloned into pCRISPRia-v2 (Addgene #84832) via BstXI-BlpI sites. After transduction with sgRNA lentivirus, K562 cells (ATCC CCL-243) were selected with 2 μg/mL puromycin (Gibco A1113802). Knockdown of target genes was assessed by RT-qPCR using PerfeCTa SYBR Green SuperMix (QuantaBio 95054-500) per the manufacturer's instructions. HPRT1 was used as endogenous control.

## RNA isolation
Total RNA for RNA-seq and RT-qPCR was isolated using the Zymo QuickRNA isolation kit (Zymo Research R1054) with in-column DNase treatment per the manufacturer's protocol.

## RNA treatment with α-amanitin
K562 (ATCC CCL-243) and K562 TAF15 knockdown cell lines were seeded at 1 M/1 mL density in 2 replicates. Cells were infected with 10 μg/mL α-amanitin(Sigma-Aldrich A2263) for 8-9 h prior to total RNA extractions. Total RNA for downstream RNA-seq was isolated using a Zymo QuickRNA Microprep isolation kit (Zymo Research R1050) with in-column DNase treatment per the manufacturer's protocol.

## RNA-seq
RNA-seq libraries were prepared using SMARTer Stranded Total RNA-Seq Kit v2 - Pico Input Mammalian (Takara 634411), and sequenced on Illumina NovaSeq 6000 instrument, in a PE150 (paired end 150 cycles) setting, at Novogene Corporation.

## Ribosome profiling
Ribosome profiling was performed as previously described[82]. Briefly, approximately $10 \times 10^6$ cells were lysed in ice-cold polysome buffer (20 mM Tris pH 7.6, 150 mM NaCl, 5 mM MgCl$_2$, 1 mM DTT (Scientific Laboratory Supplies Ltd NAT1068), 100 μg/mL cycloheximide) supplemented with 1% v/v Triton X-100 and 25 U/mL Turbo DNase (Thermo Fisher Scientific AM2238). The lysates were triturated through a 27 G needle and cleared for 10 min at 21,000 × g at 4 °C. The RNA concentration in the lysates was determined with the Qubit RNA HS kit (Thermo Fisher Q32852). Lysate corresponding to 30 μg RNA was diluted to 200 μl in polysome buffer and digested with 1.5 μl RNaseI (Epicenter VWR International 101228-268) for 45 min at room temperature. The RNaseI was then quenched by 10 μl SUPERaseIN (Thermo Fisher Scientific AM2696).

Monosomes were isolated using MicroSpin S-400 HR (Cytiva) columns, pre-equilibrated with 3 mL polysome buffer per column. 100 μl digested lysate was loaded per column (two columns were used per 200 μl sample) and centrifuged for 2 min at 600 × g. The RNA from the flow-through was isolated using the Zymo RNA Clean and Concentrator-25 kit (Zymo Research R1017). In parallel, total RNA from undigested lysates was isolated using the same kit.

Ribosome-protected footprints (RPFs) were gel-purified from 15% TBE-Urea gels (Life Technologies EC6875BOX) as 17–35 nt fragments. RPFs were end-repaired using T4 PNK (NEB M0201S), and pre-adenylated barcoded linkers were ligated to the RPFs using T4 Rnl2(tr) K227Q (NEB M0351S). Unligated linkers were removed from the reaction by yeast 5'-deadenylase (NEB M0331S) and RecJ nuclease (NEB M0264S) treatment. RPFs ligated to barcoded linkers were pooled, and rRNA-depletion was performed using riboPOOLs (siTOOLs) as per the manufacturer's recommendations. Linker-ligated RPFs were reverse transcribed with ProtoScript II RT (NEB M0368S) and gel-purified from 15% TBE-Urea gels. cDNA was then circularized with CircLigase II (Epicentre) and used for library PCR. First, a small-scale library PCR was run supplemented with 1X SYBR Green and 1X ROX (Thermo Fisher Scientific K0221) in a qPCR instrument. Then, a larger scale library PCR was run in a conventional PCR instrument, performing a number of cycles that resulted in ½ maximum signal intensity during qPCR. Library PCR was gel-purified from 8% TBE gels and sequenced on a SE50 run on an Illumina HiSeq4000 instrument at the UCSF Center for Advanced Technologies.

## ATAC-seq
The assay for transposase-accessible chromatin using sequencing (ATAC-seq) was performed according to the optimized Omni-ATAC protocol[83,84]. Briefly, samples containing 50,000 cells as input were pelleted, lysed, washed, and re-pelleted using the lysis and wash buffers specified in the Omni-ATAC protocol. A transposition mix containing Tn5 was then added to the samples, and the transposition reaction was carried out for 30 min at 37 °C in a thermomixer with 1000 rpm mixing. After transposition, the transposed DNA was purified using the DNA Clean & Concentrator-5 Kit (Zymo Research D4014). The samples underwent two PCR steps. First, a pre-amplification was performed for 3 cycles to attach unique barcoded adapters to the transposed DNA sample. The concentration of each pre-amplified sample was quantified via qPCR using the NEBNext Library Quant Kit (New England Biolabs E7630). Afterward, samples underwent a second PCR amplification step to obtain the desired DNA concentration of 8 nM in 20 μl. DNA cleanup and qPCR quantification were performed again, and the final libraries were diluted down to

exactly 8 nM using sterile water. Samples were sequenced using paired-end 75-bp sequencing on an Illumina NextSeq sequencer.

## ChIP-qPCR

ChIP-qPCR was performed as described in ref. 85. Human chronic myelogenous leukemia K562 cells (ATCC CCL-243) were grown at 37 °C and 5% CO2 in RPMI-1640 medium supplemented with 10% FBS, glucose (2 g/L), L-glutamine (2 mM), 25 mM HEPES, penicillin (100 units/mL), streptomycin (100 μg/mL) and amphotericin B (1 μg/mL) (Gibco). 20 million cells per sample were washed with PBS (in duplicate), pelleted, and cross-linked with 1% paraformaldehyde (Fisher Scientific AC416780010) for 10 min at room temperature. Glycine (Sigma-Aldrich 9002-93-1) at a final concentration of 125 mM was added to the samples and incubated at room temperature for 5 min to quench the paraformaldehyde (Fisher Scientific AC416780010). Samples were washed with PBS, pelleted, flash-frozen, and stored at − 80. Samples were thawed, lysed in 200 μl Membrane Lysis Buffer (10 mM Tris-HCl pH 8.0, 10 mM NaCl, 0.5% IGEPAL CA-630, 1X protease inhibitors), and incubated on ice for 10 min. Samples were centrifuged at 4 °C at 2500 × g for 5 min, resuspended in 200 μl Nuclei Lysis Buffer (50 mM Tris pH 8.0, 10 mM EDTA, 0.32% SDS, 1X protease inhibitors), and incubated on ice for 10 min. 120 μl of IP Dilution Buffer (20 mM Tris-HCl pH 8.0, 2 mM EDTA, 150 mM NaCl, 1% Triton X-100, 1X protease inhibitors) was added to the samples, and the samples were sonicated using the Bioruptor UCD-200 sonicator for 7 min with 30 s on/off intervals for a total of 3 times. Samples were centrifuged at 4 °C at 21000 × g for 5 min to clear the lysate, and the supernatant containing the chromatin was stored at − 80.

230 μl IP Dilution Buffer was added to 270 μl chromatin along with 3 μg ZNF800 or QKI antibody or same- species IgG, and the samples were incubated overnight at 4 °C. The next day, the ChIP samples were spun down at 4 °C at 16000 × g for 5 min, and the supernatant was transferred onto 20 μl of washed Protein A/G beads (Fisher Scientific 88802). Samples were incubated for 2 h at 4 °C.

The ChIP material was washed once with 500 μl of cold FA lysis low salt buffer (50 mM Hepes-KOH pH 7.5, 150 mM NaCl, 2 mM EDTA, 1% Triton-X 100, 0.1% sodium deoxycholate), twice with cold NaCl high salt buffer (50 mM Hepes-KOH pH 7.5, 500 mM NaCl, 2 mM EDTA, 1% Triton-X 100, 0.1% sodium deoxycholate), once with cold LiCL buffer (100 mM Tris-HCl pH 8.0, 500 mM LiCl, 1% IGEPAL CA-630, 1% sodium deoxycholate), and twice with cold 10 mM Tris 1 mM EDTA pH 8.0. Samples were eluted in 300 μl of Proteinase K reaction mix (20 mM Tris pH 8, 300 mM NaCl, 10 mM EDTA, 5 mM EGTA, 1% SDS, 60 μg Proteinase K) and incubated at 65 °C for 1 h. The supernatant was transferred to phase lock tubes (VWR), purified via phenol-chloroform extraction, and eluted in 30 μl 10 mM Tris pH 8.0.

qPCR was performed using PerfeCTa SYBR Green SuperMix (QuantaBio) per the manufacturer's instructions. HPRT1 was used as endogenous control.

## Crosslinking and immunoprecipitation

K562 cells (ATCC CCL-243) were harvested and crosslinked with ultraviolet radiation (400 mJ/cm2). Cell lysates were then treated with high (1:3000 RNase A and 1:100 RNase I) and low dose (1:15000 RNase A and 1:500 RNase I) of RNase A (Thermo Fisher Scientific EN0531) and RNase I (Thermo Fisher Scientific EN0601) separately and combined after treatment. Antibodies to TAF15 (Thermo MA3-078, dilution according to manufacturer's recommendation) or ZC3H11A (Abcam ab241612, dilution according to manufacturer's recommendation) were first conjugated to protein A/G beads (Pierce) and then added to cell lysates to immunoprecipitate protein-RNA complex. This was followed by beads dephosphorylation, polyadenylation, and IRDye® 800CW DBCO Infrared Dye (LI-COR 929-50000) end labeling of the immunoprecipitated RNA fragments. RNA-protein complex was then resolved by SDS-PAGE and visualized on nitrocellulose membrane.

Membranes were then cut and treated with proteinase K to release RNA. We then used Takara smarter small RNA sequencing kit reagents with a custom UMI-oligo dT primer (CAAGCAGAAGACGGCATACGA GATNNNNNNNNNGTGACTGGAGTTCAGACGTGTGCTCTTCCGATCTTT TTTTTTTTTTTTT) to synthesize cDNA. Sequencing libraries were then prepared with SeqAmp DNA Polymerase (Takara 638509) and sequenced on an Illumina Hiseq 4000 sequencer.

## Immunofluorescence assay

K-562 cells were seeded and grown on Poly-D-Lysine (MP Biomedical 0215017550) coated chamber slides (SPL 30108). Cells were fixed with 4% paraformaldehyde (PFA) (Fisher Scientific AC416780010) for 5 min at room temperature, followed by permeabilization with 0.5% PBST for 10 min and blocking with 4% BSA for 1 h. Primary antibodies were diluted according to manufacturers' recommendation and incubated overnight at 4 C. Cells were then stained with a standard amount of fluorescent secondary antibody for 1 h at room temperature. Samples were then mounted in ProLong™ Gold Antifade Mountant with DAPI (Life Technologies P36941) and imaged with a (Zeiss LSM 780) confocal microscope (courtesy of the Cardiovascular Research Institute at UCSF).

## Computational tools

Reanalysis of enhanced CLIP ENCODE data. To reliably identify RNA targets of RBPs in K562 cells (ATCC CCL-243), we started with the raw eCLIP FASTQ files of 'released' K562 experiments for 120 RBPs that were available in the ENCODE database. The analysis was performed as follows: (1) the reads were preprocessed in the same way as in ref. 18 including adapter trimming with *cutadapt* (v1.18)[86], (2) preprocessed reads were mapped to the hg38 genome assembly with GENCODE v31 comprehensive annotation using *hisat2* (v.2.1.0)[87], (3) the aligned reads were deduplicated using the barcodecollapsepe.py script (https://github.com/YeoLab/eclip/tree/master/bin) as in ref. 18, (4) properly paired and uniquely mapped second reads were extracted using *samtools* (v.1.9, with -f 131 -q 60 parameters)[88], (5) gene-level read counts were obtained with *plastid* (v.0.4.8) by counting 5' ends of the reads[89], (6) analysis of specific enrichment against size-matched control experiments was performed with *edgeR* (v.3.18.1) for each RBP separately, considering only genes passing 2 cpm in at least 2 of 3 samples[90]. Reliable RNA targets of each RBP were defined as those passing 5% FDR and log$_2$(Fold Change) > 0.5, see Supplementary Data File 8. eCLIP target scores (TSs) used in datasets integration were estimated as -log$_{10}$(P)·sign(log$_2$FC) for every "RBP-gene" pair separately.

## MS data analysis (BioID2 mass spectrometry data)

Data were quantified and queried against a Uniprot human database (January 2013) using *MaxQuant* MaxLFQ command[91]. Data normalization was performed in Perseus[92] (version 1.6.2.1). For batch correction, Brent Pedersen's implementation[93] of the ComBat function from *sva* package[94] was used. The protein abundances in "experiment" (biotin +) and "control" (biotin −) samples were compared using *t* test for each protein individually.

## Perturb-seq analysis

Cell Ranger (version 3.0.1, 10X Genomics) with default parameters was used to align reads and generate digital expression matrices from single-cell sequencing data. To assign cell genotypes, a bwa ref. 95 database was created containing all guide sequences present in the library using the bwa index command. The barcode-enrichment libraries were mapped to this database to establish the guide identities; to detect the cell barcodes, the barcode correction scheme used in Cell Ranger was used (the mapping of uncorrected to corrected barcodes was extracted from Cell Ranger analysis run of the whole transcriptome libraries; this mapping was then applied to the reads of barcode-enrichment libraries). UMI correction was performed by

merging the UMIs within the hamming distance of 1 from each other. For each UMI, the guide assignment was done by choosing the guide sequence most represented among the reads containing the given UMI. To make the final assignment of a guide to cell barcodes, we only considered the barcodes that were represented by at least 5 different UMIs, with > 80% of UMIs representing the same guide.

Data filtering was performed using *scanpy* package[96]. Data were denoised using a modification of *scvi* autoencoder[97] with loss function penalizing for the similarity between cells having different RBPs knocked down. The distance between transcriptome profiles of individual RBP knockdowns was calculated by applying the *t* test to individual gene counts across the cells that were assigned the respective guide sequence.

### Dataset integration
The functional similarity of RBPs was estimated by joint analysis of eCLIP, BioID, and Perturb-seq data (Fig. 1 and Supplementary Fig. S1). First, TS Z-scores were calculated for every gene across RBPs separately for each type of experimental data (eCLIP, BioID, or Perturb-seq) in the same way as preys of the BioID data, see above and Supplementary Fig. S1(1). Next, cosine distance was computed for all 7140 pairs of different RBPs, followed by ranking and calculation of empirical *p*-values defined as a fraction of RBP pairs with the cosine distance less than the score of the tested pair, see Supplementary Fig. S1(2). The empirical *p*-values were aggregated with *logitp* function from the *metap* R package (v.1.4)[98], see Supplementary Figs. S1(3, 4), for 4005 RBP-RBP pairs (90 proteins in total) with at least 2 out of 3 available data types. Heatmap.2 functions of the *gplots* R package (v.3.1.1) with cosine distance and Ward's (ward.D2) clusterization were used to generate the integration heatmap shown in Fig. 2.

STRING-based RBP interaction heatmap was generated using protein links' combined scores (STRING v.11.5) and the same clustering method as in the dataset integration procedure[19]. To test the overlap between the integrated interaction map and the external databases, we also downloaded significant protein-protein interactions from OpenCell[5] and hu.MAP[22] databases. With these data, we estimated the fractions of the interactions found in STRING with the combined score over 400 (medium confidence STRING interactions), in OpenCell, in hu.MAP with a score over 0.02 (medium confidence hu.MAP interactions) and in Zanzoni et al. with at least 150 complexes shared between RBPs, among the RBP-RBP pairs with the integrated distance passing a selected quantile threshold (Supplementary Fig. S2E). We also considered the fractions of OpenCell- and hu.MAP-based interactions among the pairs not included in STRING medium confidence interactions. To estimate the significance of the intersection, we performed the same procedure with $10^4$ random shuffles of the IRIM. Finally, the empirical *p*-values were estimated and corrected for multiple testing using Benjamini-Hochberg (FDR) adjustment. To estimate the consistency of the results depending on the datasets used for distance integration, we additionally performed the procedure described above using distances integrated from eCLIP and Perturb-seq (2278 RBP-RBP pairs with both datasets available, *p*-value < $10^{-4}$), BioID and Perturb-seq (378 pairs, *p*-value = 0.028), BioID and eCLIP (1225 pairs, *p*-value < $10^{-4}$, Supplementary Fig. S2E).

To evaluate the stability of protein-protein interactions within the IRIM, columns in the $90 \times 90$ matrix were shuffled at varying fractions of columns (1, 5, 10, 25, and 50%) to observe alterations in inter-RBP distances and matrix topology. The shuffling involved the calculation of cosine distances from each of the original 90 RBP's distance vectors to the respective vectors in the partially shuffled matrix, focusing on the minimal, median, and maximal distances to other RBPs. This procedure was repeated 10 times, generating 900 estimates for each group and percentage of shuffled columns, ensuring a comprehensive analysis of distance variations and topological alterations. To compare the inter-RBP distances stabilities of the IRIM and STRING, OpenCell or

hu.MAP, the same procedure was applied to the respective binary interaction matrices 10 times for each shuffling percent. For STRING and hu.MAP, interactions were considered valid if the STRING combined score was > 400 or hu.MAP score was > 0.02, respectively, and for RBPs, protein-protein interactions were assumed if the distance was within the < 25% quantile of the IRIM. Moreover, 90 RBPs were randomly chosen from the STRING, OpenCell, and hu.MAP interaction matrices before shuffling to make their sizes comparable to the IRIM. This procedure was repeated 5 times.

### Transcript types enrichment analysis of RBP RNA targets
A joint set of 22471 genes detected at 2 counts per million (cpm) in at least two samples of one eCLIP experiment was used as the background for further analysis. RBPs preferences to bind RNAs of a particular type were assessed using a one-sided Fisher's exact test. The following types of RNAs were selected based on GENCODE annotation: miRNA, lncRNA, protein_coding, snRNA, snoRNA, and rRNA. For each RBP separately, the *p*-values were adjusted for multiple testing using FDR correction for the number of tested RNA types. Visualization of the eCLIP, RNA-seq, and ATAC-seq profiles generated using *bedtools genomecov* (v.2.27.1) was performed with *svist4get* (v.1.2.24)[99,100].

### Functional annotation of RBPs
To annotate the RBPs based on prey identified in BioID experiments, target scores (TSs) were estimated as -log$_{10}$(P)·sign(log$_2$FoldChange) for every bait-prey pair separately. Next, for each prey, TSs were converted to Z-scores by estimating mean and average across baits. The preys were ranked by Z-scores, and the Fgsea R package (v.1.12.0) was applied to perform gene set enrichment analysis with 100000 permutations and three GO terms annotation sets (BP, MF, and CC, each taken separately)[57]. The annotation sets were generated with the go.gsets function of *gage* R package (v.2.36.0)[101]. Lists of 2865 Entrez ids of preys were used in *fgsea* analysis for each RBP of the total set of fifty. GO terms with NES > 2 for at least one RBP were considered when plotting Fig. 3 and Supplementary Fig. S3 (related GO terms were merged manually), negative NES were zeroed for clarity and easier interpretability of the consequent clusterization, see complete data in Supplementary Data File 3). Ward.D2 clusterization along with cosine distance (1 - cosine similarity) were used to generate the heatmaps using the heatmap.2 function of the *gplots* R package (v.3.1.1)[102].

To check the consistency between predicted and known RBP annotations, the same procedure was performed excluding the Z-scoring step to avoid penalizing common generic GO terms e.g., "organelle", "cell", etc. The resulting GSEA *p*-values and NESs were used to calculate the <RBP, GO term> scores as -log$_{10}$(P)·sign(log$_2$FoldChange) for each RBP and GO term separately. RBPs' "true" annotations were extracted from the same GO BP, CC, or MF annotation set as used in GSEA. Finally, all data were merged to generate the ROC curve with *PRROC* (v.1.3.1) roc.curve function[103].

### Alternative splicing analysis
We used MISO[104] for alternative splicing analysis, as this tool is known for its consistent performance and wide use[105]. Specifically, RNA-seq data was processed as follows: (1) to fulfill MISO requirements (see below), the reads obtained with different sequencing lengths were truncated to 75 bps with *cutadapt* (v.2.10) -l option, (2) the truncated reads were mapped to the human hg38 genome assembly with GENCODE v38 comprehensive gene annotation using *STAR* (v.2.7.9) with options --outFilterScoreMinOverLread and --outFilterMatchNminOverLread both set to 0.25[106], (3) non-unique alignments were filtered, and the replicates were merged, (4) the insert size distribution was estimated for each merged bam file separately using pe_utils −compute-insert-len from *MISO* (v.0.5.4), constitutive exons were retrieved using exon_utils with --get-const-exons and --min-exon-size 1000[104], (5) alternative splicing events were identified using miso --run with --read-len set to 75 and

--paired-end set to the previously estimated insert size parameters. Finally, only cases with non-zero numbers of exclusion and inclusion read, and the sum of these reads ≥10 in at least one sample is left and shown in Fig. 4.

## Ribosome profiling analysis

To process the reads, the Ribo-seq reads were first trimmed using *cutadapt* (v2.3) to remove the linker sequence AGATCGGAAGAGCAC. The fastx_barcode_splitter script from the *Fastx* toolkit was then used to split the samples based on their barcodes. Since the reads contain unique molecular identifiers (UMIs), they were collapsed to retain only unique reads. The UMIs were then removed from the beginning and end of each read (2 and 5 Ns, respectively) and appended to the name of each read. *Bowtie2* (v2.3.5) was then used to remove reads that map to ribosomal RNAs and tRNAs, and the retained reads were then aligned to mRNAs (we used the isoform with the longest coding sequence for each gene as the representative). Subsequent to alignment, *umitools* (v0.3.3) was used to deduplicate reads.

The quality check and downstream processing of the processed reads was performed using *Ribolog* v0.0.0.9[14]. To distinguish stalling peaks from stochastic sequencing artifacts, we followed a multi-step procedure. We calculated P-site offsets and identified the codon at the ribosomal A-site for each RPF read using the *riboWaltz* package. A loess smoother was used to de-noise codon-wise RPF counts. The loess span parameter varied depending on the transcript length and allowed borrowing information from ~5 codons on either side of the A-site. We calculated an excess ratio at each codon position by dividing the loess-smoothed count by the transcript's background translation level (median of no-zero loess-smoothed counts). After median normalization of the corrected counts and removal of transcripts with 0 counts, the ribosome occupancy testing was performed using logistic regression in *Ribolog*.

## ATAC-seq analysis

ENCODE ATAC-seq pipeline[107] with default parameters was used for sequencing data processing and analysis. The differentially accessible peaks were identified with the DESeq2 package[108] and annotated with the *ChIPseeker* package[109]. To perform a comparison against published ChIP-Seq data, the processed ChIP-exo results were downloaded from GEO (GSE151287)[68]. The data consisted of bed files containing 33 and 181 QKI peaks (two replicates) and a bigWig file with ZNF800 ChIP-exo signal (no ChIP-exo peaks were reported for ZNF800). In total, 234564 and 222350 ATAC-seq peaks for QKI and ZNF800, respectively, had coverage of at least 10 reads in more than one replicate and were used in the following tests. For QKI, the bed files with ChIP-exo peaks were merged, transferred to the hg38 genome assembly with UCSC *liftOver* and the numbers of differentially accessible (or not differentially accessible) QKI-KD ATAC-seq peaks that intersect (or do not intersect) ChIP-exo peaks were calculated using bedtools intersect (v.2.26.0)[99,110] followed by a one-sided ('greater') Fisher's exact test on 2 × 2 contingency table. For ZNF800, bigWig files were converted to bed using UCSC bigWigTo-Wig (v.377) and wig2bed from BEDOPS (v.2.4.38)[111,112], followed by UCSC *liftOver* to the hg38 genome assembly. The resulting regions were intersected with differentially accessible and not differentially accessible ZNF800-KD ATAC-seq peaks using bedtools intersect, followed by a comparison of ChIP-exo signal distribution in these two sets using non-parametric Mann-Whitney U test.

## Mass Spectrometry data analysis (TAF15 KD proteomic quantification)

Quantitative analysis of the TMT experiments was performed simultaneously with protein identification using *Proteome Discoverer 2.5* software. The precursor and fragment ion mass tolerances were set to 10 ppm, 0.6 Da, respectively), the enzyme was Trypsin with a maximum of 2 missed cleavages, and the UniProt Human proteome FASTA file and common contaminant FASTA file was used in SEQUEST searches. The impurity correction factors obtained from Thermo Fisher Scientific for each kit were included in the search and quantification. The following settings were used to search the data; dynamic modifications; Oxidation / + 15.995 Da (M), Deamidated / + 0.984 Da (N, Q), Acetylation /+ 42.011 Da (N-terminus), and static modifications of TMT6plex / + 229.163 Da (N-Terminus, K), MMTS / + 45.988 Da (C).

*Scaffold Q +* (version Scaffold_5.0.1, Proteome Software Inc., Portland, OR) was used to quantitate TMT Based Quantitation peptide and protein identifications. Peptide identifications were accepted if they could be established at greater than 78.0% probability to achieve an FDR less than 1.0% by the Percolator posterior error probability calculation[113]. Protein identifications were accepted if they could be established at greater than 5.0% probability to achieve an FDR less than 1.0% and contained at least 1 identified peptide. Protein probabilities were assigned by the Protein Prophet algorithm[114]. Proteins that contained similar peptides and could not be differentiated based on MS/MS analysis alone were grouped to satisfy the principles of parsimony. Proteins sharing significant peptide evidence were grouped into clusters. Channels were corrected by the matrix [0.000,0.000,0.931, 0.0689,0.000]; [0.000,0.000,0.933,0.0672,0.000]; [0.000,0.00750, 0.931,0.0619,0.000]; [0.000,0.0113,0.929,0.0593,0.000]; [0.000, 0.0121,0.934,0.0532,0.000934]; [0.000,0.0148,0.923,0.0499, 0.0120]; [0.000,0.0251,0.931,0.0438,0.000]; [0.000,0.0206,0.936, 0.0431,0.000]; [0.000,0.0291,0.937,0.0337,0.000]; [0.000,0.0776, 0.892,0.0303,0.000] in all samples according to the algorithm described in i-Tracker[115]. Normalization was performed iteratively (across samples and spectra) on intensities, as described in Statistical Analysis of Relative Labeled Mass Spectrometry Data from Complex Samples Using ANOVA[116]. Means were used for averaging. Spectra data were log-transformed, pruned of those matched to multiple proteins, and weighted by an adaptive intensity weighting algorithm. Of 22889 spectra in the experiment at the given thresholds, 20372 (89%) were included in quantitation. Differentially expressed proteins were determined by applying *t* test with an unadjusted significance level of *p*-value < 0.05, corrected by Benjamini-Hochberg.

## Statistics & reproducibility

Statistical parameters are reported in the figures and figure legends, including the definitions and experimental measures depicted either as bar charts representing mean and dot plots representing exact values or as boxplots representing median, 25th and 75th percentile (boxes), and 5% and 95% confidence intervals (error bars). For the BioID-based RBP annotation procedure, statistical significance is indicated by asterisks * if GSEA FDR adjusted *p*-value < 0.05. Pairwise comparisons of qPCR results and log-transformed MS intensity ratios were performed using a one-sided *t* test (for testing alternative splicing) or Wilcoxon rank sum test (for testing protein levels and mRNA relative stability). Exact *p*-values are depicted above the corresponding bar charts. For TAF15 mRNA target enrichment analysis, GSEA statistics, including *p*-values and enrichment scores, are depicted in the figure. To test the intersection of different TAF15 regulons, *p*-values were calculated using one-sided Fisher's exact tests with the statistical significance indicated by asterisks *, *p*-value < 0.05, **, *p*-value < $10^{-5}$. Pairwise comparisons of the QKI and ZNF800 target expression level and chromatin accessibility were performed using a one-sided Wilcoxon rank sum test with exact *p*-values depicted above the boxplots.

## Reporting summary

Further information on research design is available in the Nature Portfolio Reporting Summary linked to this article.

**Article** https://doi.org/10.1038/s41467-024-52215-7

## Data availability

All the sequencing data has been deposited at Gene Expression Omnibus ([GSE225809]) and are publicly available as of the date of publication. The processed data has been deposited to Zenodo (identifier [11556393]). The list of bona fide BioID protein pairs has been deposited to IMEx (identifier [IM-30059]). The mass spectrometry proteomics data have been deposited to the ProteomeXchange Consortium via the PRIDE partner repository with the dataset identifier [PXD041608]. Source data are provided in this paper.

## Code availability

All the original code has been deposited to [GitHub](https://github.com/goodarzilab/RBP_modules) and [Zenodo](https://zenodo.org/records/10498278) and is publicly available as of the date of publication. The RBP Browser is publicly available at [https://goodarzilab.shinyapps.io/RBP-Browser/](https://goodarzilab.shinyapps.io/RBP-Browser/).

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

## Acknowledgements

The authors thank Artemii Bakulin, Heather Karner, and Ilia Vorontsov for helpful discussions. D.M. was supported by an M.D. fellowship from the Boehringer Ingelheim Fonds. M.D. and F.K.M. were supported by the United Kingdom's Medical Research Council (MRC) grants MR/P009417/1 and MR/W001500/1.

## Author contributions

M.K., F.K.M., and H.G. designed the study. M.K and J.Y. performed Perturb-seq experiments. M.K., A.N., F.T., A.D., R.B., and M.D. performed proximity labeling experiments. M.K., F.T., A.D., and D.M. performed western blotting experiments. A.B. and I.K. performed a re-analysis of ENCODE eCLIP data. M.K., A.B., and I.K. performed the dataset integration. M.K., A.B., and I.K. performed the RBP functional annotation. M.K. performed CRISPRi knockdown experiments. M.K. and B.C. performed RNA-seq experiments. M.K., H.M., and R.C. performed ATAC-seq experiments. K.G. and H.G. performed ChIP-qPCR experiments. D.M. and H.G. performed qPCR experiments. T.J. and B.C. performed α-amanitin treatment experiments. A.N. performed ribosome profiling experiments. M.K., A.B., S.M., V.S., C.C., and H.G. performed data analysis. V.S. built the RBP Browser app. S.B.L. and S.Z. performed the immunofluorescence experiments. S.Z. performed CLIP-seq experiments. M.K., A.B., I.K., and H.G. wrote the manuscript with input from all authors.

## Competing interests

The authors declare no competing interests.
