## [Peer Review File · Nature Communications]

Systematic Identification of Post-Transcriptional Regulatory ModulesReviewer #1 (Remarks to the Author):

In this manuscript, Khoroshkin and colleagues aim at the identification of post-transcriptional regulatory modules for a relatively small subset of canonical RNA-binding proteins (RBPs).

To achieve this, they generated a regulatory map (IRIM) by integrating three distinct datasets, namely in vivo BioID-based protein proximity mapping, CRISPRi experiments and protein-RNA interactions from eCLIP assays. Only the latter was not generated in this manuscript.

The resulting map allowed the authors to characterize several regulatory modules, which broadly confirmed previous knowledge on the functional roles of the studied RBPs, but also pinpointed additional ones for some RBPs. For instance, the integrated regulatory map suggested that several RBPs can participate in different stages of RNA processing or, interestingly, play a role in the regulation of transcription. The authors selected 4 of these RBPs and performed additional experiments (e.g., RNA-seq, Ribo-Seq, ATAC-seq and ChIP-qPCR) to confirm the functional roles inferred from the IRIM.

Overall, the authors have generated an impressive amount of data that provide very interesting insights for the characterization and the understanding of post-transcriptional regulatory networks. This work represents a valuable resource for the community.

Nevertheless, I have several concerns to raise on different aspects of this work that should be addressed.

First of all, I believe that the authors should clarify the conceptual biology framework of their work.

For instance, reading the manuscript, it was not clear to me what the authors' definition of RNA regulon.

Previous established literature on this topic, which was partially overlooked by the authors in the introduction (e.g., PMID: 17572691 PMID: 18959479 PMID: 25436546), define an RNA regulon as an ensemble of functionally-related transcripts whose fate is coordinately regulated by RBPs and/or other regulatory molecules such microRNAs.

In the introduction, the authors underline that the simplistic model of "one RBP – one function – one regulon" is not sufficient to capture the complexity of post-transcriptional regulation (statement to which I agree), and suggest that a "one RBP – many functions – many regulons" is more appropriate.

To support this, they mention the example of MBNL1 that regulates splicing in the nucleus while modulating stability and localization of target RNAs in the cytoplasm. However, this does not tell us whether MBNL1 regulating the same mRNAs both in the nucleus and the cytoplasm or whether these mRNAs involved in the same cellular function(s), which is essential for the definition of a RNA regulon.

Later in the manuscript, while dissecting TAF15 role in splicing and RNA stability, they point out that RBP is involved in three different, although overlapping, regulons: alternative splicing, differential translation and stability (I have additional comments on this point, see below). However, it is not clear whether the regulated transcripts are involved in similar or different

cellular functions. It seems to me that, in this case, the authors defined the regulon based on the role of the RBP rather than grounding it on the function(s) of the targets.

This brings me to another general concern regarding the definition of multifunctionality that the authors employ throughout the manuscript.

Multifunctional proteins, known also with a stricter definition as moonlighting proteins, are proteins that perform multiple distinct functions in the cell (e.g., PMID: 10087914, PMID: 20144902, PMID: 22696112, PMID: 26220711). Many RBPs can be defined as such (see aconitase for example), but they are often non-conventional RBPs (e.g.: PMID: 20554447, PMID: 26520658).

The authors seem to define multifunctional RBPs as they participate in different steps of RNA fate either in the nucleus and/or the cytoplasm. Given the established definition of multifunctionality, this should not be the case since they are always acting as RNA binders in different steps of the same process. Therefore, authors' definition looks to me closer to a "functional pleiotropy" of RBPs (i.e. doing the same task in different context) rather than protein multifunctionality or protein moonlighting.

When referring to their BioID dataset, the authors often mention "physical interactions" or "protein interactions". However, as correctly described in the abstract, BioID proximity labelling allows to define in vivo protein neighborhoods of the proteins of interest and does not allow to infer any physical contact among these proteins. The authors should be consistent with this terminology in the manuscript.

Overall, these concepts and definitions should be clarified, in particular in the introduction and discussion, and rephrased throughout the manuscript when needed.

In the discussion, understandably, the authors underline the relevance of their work and what are the next steps to take for improving our knowledge of post-transcriptional regulatory networks. Authors should also discuss any limitations that their study may have (e.g., number of studied RBPs, availability of relevant data such eCLIP, etc.).

Besides these broad concerns, I have some more specific comments and suggestions:

a)

The authors should consider mentioning in the introduction previous works that aimed at the characterization of post-transcriptional regulatory networks either computationally (e.g., PMID: 30867517 and PMID: 31733516) or through experimental approaches (e.g. PMID: 34133714, PMID: 37070168). Contrast with these previous works could be also beneficial for enriching the discussion section.

In particular, the dataset from Lang et al. could be used to support the IRIM clustering (e.g., are RBPs in the same clusters also are in physical contact?).

This analysis could be extended to additional resources, like the Human Reference Interactome (HuRI, <http://www.interactome-atlas.org/>), the Human Protein Complex Map (hu.MAP 2.0,

<http://humap2.proteincomplexes.org/>), which includes BioPlex, and also to, and maybe more importantly, the Ribonucleoprotein Complex Map (rna.MAP, <http://rna.proteincomplexes.org/>) that collect human protein complexes containing RNAs.

The comparison to these protein interaction datasets could be beneficial not only for assessing the IRIM, but also to provide orthogonal “physical” support to the BioID data.

All this can be done in the same statistical framework that the authors used in the manuscript.

b)

The IRIM was generated through late multimodal data integration and hard clustering based on cosine distance to identify putative regulatory modules. The authors summarize the proposed approach in Figure S1. This figure should be improved by provide detailed explanation of the terms used in the equations.

In addition, it is not clear to me how the modules were functionally annotated. Was it done by labelling each cluster with the most frequent annotation among the RBPs or through enrichment analysis or consensus knowledge? Are these Gene Ontology annotations? Are the authors taking into account the nature of the evidence supporting the association between a given annotation and an RBP (i.e., including IEA annotations)? Please, explain better this point as this is key for the rationale of the work presented in the manuscript.

Related to this, in the lower panels of Figure 2, the authors present some illustrative examples of identified functional partners based on their integrated or dataset-specific cosine distance. These panels highlight that close RBP also shared a common function with the RBP interest. How many cases like these are found in the IRIM? It would important to provide swarm plots for all the RBP in the IRIM as supplementary figure by highlighting those RBP with common function and see whether and how this correlates with distance.

In Figure S2D, the authors compared the IRIM with an analogous matrix built using STRING functional interactions. Below the heatmap is reported the color scale for STRING confidence score, but in the legend the authors refer to a distance metric. Please, clarify.

Given the hard clustering method used by authors, RBPs in the IRIM could belong to only one cluster (“one RBP – one function – one regulon”). To spot RBPs that could participate in different functions, the authors indeed looked for off-diagonal signals. Did the authors try to apply different “soft” clustering approaches, like fuzzy clustering, to check whether some RBPs could be multi-clustered? I am asking this just for sake of comparison.

c)

The BioID proximity data has been also used by the authors to go beyond the RBP-RBP relationships and exploit the protein neighborhoods to gain further functional insights. As mentioned earlier in my report, I would perform an extensive comparison with additional protein interaction datasets to add a physical contact layer to the identified neighborhoods.

I think that this dataset is not fully exploited and the authors could perform additional scrutiny.

For instance, among RBP neighboring proteins, how many are known RBPs, how many are non-conventional RBP? Do RBPs share neighbors? Is there any particular class of proteins that is present among all, or in some neighborhoods?

In this part, the authors use in several occasions the “RBP-pathway” term when describing the functions associated to RBPs and/or regulatory modules. However, they use Gene Ontology terms from the different branches (BP, MF and CC) that are not necessarily “biological pathways”. Please, pick a more appropriate term.

The BioID data should be provided in a more reader-friendly in a tabular format with bait-prey pairs that can be considered as “bona fide” associations.

Likewise, for the results of GSEA (S12, S13, S14), a tabular format visualization of the data should be provided for easy exploration (i.e., RBP-annotation pairs, ES, NES, nominal p-value, FDR, $-\log_{10}(\text{FDR})$, list of neighboring proteins associated to the given annotation).

Out of curiosity, did the authors tested also for significantly depleted annotations in their GSEA analysis? This could provide some interesting insights on which functions are not in the proximity of a given RBPs.

Regarding Figure 3B, the red over blue is not legible, even more for the asterisks indicating the relatively few significantly enriched BP annotations. Suggestion: try with complementary colors or a combination that could be clearly more distinguishable for color-blind people.

Finally, I strongly recommend the authors to submit their BioID protein proximity data, or at least the subset that they consider as “detected protein neighborhoods” of the bait RBPs, to one of the IMEx consortium databases (for instructions, check <http://www.imexconsortium.org/submit-your-data/>).

d)

The authors focused their attention on a handful of RBPs (ZCH3H11A, TAF15, QKI and ZNF800) for further validations of their functional roles inferred from the IRIM. In particular, they sought to verify the role of ZC3H11A and TAF15 in splicing regulation. For both RBPs there are some evidences of their “putative” involvement in such process. Especially for TAF15, previous works, cited also by the authors, proved a role in splicing, although in a mouse brain model. Indeed, in this manuscript the number of ASEs identified upon TAF15 knock-down (i.e., 190) is comparable to the one detected in Kapeli et al. (i.e. 187). Despite the fact that the experimental models are different, can the authors observe any overlap between the two experiments? In any case, the outcome of the TAF15 ASE experiment is confirmatory. Therefore, this should be reported as such throughout the manuscript.

It would be interesting as well to have some descriptive details on the different ASEs detected (intron retention, exon skipping, etc.) for both RBPs and their relative proportions.

In Figure 5E, the authors show the overlap among the set of genes whose transcripts are potentially regulated by TAF15 at three different steps of the RNA fate process. The statistical significance of these overlap is assessed with a one-sided Fisher’s exact test. It seems that only

two pair sets show a significant overlap. However, what I get from the figure is that when testing the overlap, the authors forgot to include the 13 genes that are present in all three sets. Is this the case? If yes, please redo the test, otherwise clarify this in the figure legend and/or manuscript. Please also provide exact p-value from the Fisher's test for all the pairs and adjust them for multiple testing correction.

The PCA plots in Figures S4C and S5E show that two replicates (TAF15-KD and WT-untreated, respectively) are not relatively close to each other. Can the authors comment on that?

Finally, I strongly recommend the authors to provide, beside the raw data deposited in GEO, the processed analyzed data (ASEs, Ribo-Seq, ATAC-seq, etc.) as supplementary material or as an archive in data sharing repositories such as Zenodo along with a proper documentation.

e)

I downloaded and tested the code associated with the manuscript. The documentation is sufficient for running the provided scripts. Nevertheless, the authors should provide more detailed comments within the R scripts. In addition, the script should also take care of installing automatically the required libraries if they are missing on the user operating system (OS). However, as the authors correctly state, the installation of the libraries may depend on the OS present on the user computer. As a matter of fact, I was not able to install some libraries and run the scripts because I do not have a recent version of R on my computer. Thus, the optimal solution is to provide the analysis script in a dockerized environment with all the dependencies installed.

Additional comments:

* I suggest the authors to organize the methods' section in a way that it is consistent with the flow of the manuscript in the result section.

* The supplementary tables are poorly described and annotated. Please, provide more details descriptions within the data files. Readers should be able to understand what the tables contains without going back and forth between this material and the manuscript.

* The STRING database provide information about protein functional interactions that are not necessarily based only on physical interaction data. Sometime the authors forget to mention this detail in the manuscript.

Reviewer #1 (Remarks on code availability):

I downloaded and tested the code associated with the manuscript. The documentation is sufficient for running the provided scripts. Nevertheless, the authors should provide more detailed comments within the R scripts. In addition, the script should also take care of installing automatically the required libraries if they are missing on the user operating system (OS). However, as the authors correctly state, the installation of the libraries may depend on the OS present on the user computer. As a matter of fact, I was not able to install some libraries and

run the scripts because I do not have a recent version of R on my computer. Thus, the optimal solution is to provide the analysis script in a dockerized environment with all the dependencies installed.

Reviewer #2 (Remarks to the Author):

- What are the noteworthy results?

Compared to TF interactions that regulate gene expression, the post-transcriptional RBP interactions are still not quite clear. In this study, the authors constructed a map of RBP-RBP interactions by integrating data from physical and functional RBP interactions, and highlighted the value of this RBP interaction map in revealing the functions of some RBPs.

- Will the work be of significance to the field and related fields? How does it compare to the established literature? If the work is not original, please provide relevant references.

Although most of the important references on this topic have been cited, they still missed some critical ones – such as Xiao et al, Cell (PMID 31251911) and Li et al, Genome Biology (PMID 28886744).

It is also important to show the overlap between the RBP modules predicted by this study and those revealed by previous studies.

- Does the work support the conclusions and claims, or is additional evidence needed?

The authors integrated three types of data – proximity-based, target-based and function-based data – to predict the RBP modules. Given that the RBPs can interact with each other at both the levels of transcription and post-transcription, the results from this study cannot distinguish whether the RBP modules may function at which level. Their BioID experiments can capture both RNA-bound and chromatin-bound RBPs; the data from the Pertub-seq experiments should be treated as an integrated readout of RBP perturbation from both the levels of transcription and post-transcription; while the eCLIP data can only support RBP interactions at the RNA level.

The authors claim that the RBP modules identified by the integrated analysis are “post-transcriptional regulatory modules” (Figure 2). I suggest the authors double check the ChIP-seq data from Xiao et al, Cell (PMID 31251911) to exclude the false positive ones or analyze the RNA-associated and chromatin-associated RBP modules separately.

- Are there any flaws in the data analysis, interpretation and conclusions? Do these prohibit publication or require revision?

I would suggest the authors to analyze the motif component of the RBP modules. I’m wondering whether the RBPs with the same or similar motif component or not would be in the same RBP module.

- Is the methodology sound? Does the work meet the expected standards in your field?

In the main figure, the authors should show the overlap of their RBP pairs detected from BioID and predicted RBP modules with the data from some popular PPI databases. The authors put these results in the supplementary files, but I think these results are critical for evaluating the confidence of the predicted RBP modules.

- Is there enough detail provided in the methods for the work to be reproduced?
The Methods section was well written and easy to follow.

Some writing issues – for example, “i.e.” should be “i.e.,” (Line 147).

Reviewer #2 (Remarks on code availability):

The authors provided a good Readme file, which is helpful for running the codes.

Reviewer #3 (Remarks to the Author):

This study used a systems biology approach to identify the post-transcriptional regulatory modules between RNA-binding proteins (RBPs) and target regulons. A comprehensive dataset has been generated in cell lines using RNA-seq, CRISPRi knockdown, BioID2-RBP fusion cell line, mass spectrometry, Perturb-seq, ribosome profiling, ATAC-seq, and ChIP-qPCR etc. Together with public data sources like CLIP ENCODE, post-transcriptional regulatory modules (as shown in Fig. 2) were identified as clusters in a heatmap. Functional annotation of RBPs was also conducted based on BioID-mediated proximity labeling. Specific examples such as ZC3H11A and TAF15 were also presented to illustrate the capability of RBPs regulating multiple independent regulons via distinct pathways. Data analytics were described with necessary details given (e.g., tool names and version numbers). Overall, this study presented a solid piece of work on unveiling regulatory relationships between RBPs and target regulons, with novel scientific insights gained by digging into selected examples. There are a few minor concerns/suggestions though:

1. The choice of analytic pipeline tools was not clearly justified. At least some of the tools used are not of the state-of-the-art or not of the best performance (e.g, RNA-seq analysis tools). One sentence may be added to give some justification.
2. Instead of heatmap as well as biclustering algorithm, the relationship between RBP units and target regulons might be better depicted using graphs/networks. In that case, the modules obtained will correspond to graph/network communities, and the interactions between RBP units and regulons are thus the interactions between communities (a multiple-to-multiple relationship).
3. A clearer definition of RBP unit is expected for readers to understand the results. For instance, is it completely data-driven or a mixture of data and knowledge?

Reviewer #3 (Remarks on code availability):

Source codes were provided for data integration and BioID annotation. The rest of computing codes for, e.g., RNA-seq analysis, was not provided, which is acceptable as standard scripts are publicly available at many places.

Reviewer #1:

1.1 General comments

In this manuscript, Khoroshkin and colleagues aim at the identification of post-transcriptional regulatory modules for a relatively small subset of canonical RNA-binding proteins (RBPs).

To achieve this, they generated a regulatory map (IRIM) by integrating three distinct datasets, namely in vivo BioID-based protein proximity mapping, CRISPRi experiments and protein-RNA interactions from eCLIP assays. Only the latter was not generated in this manuscript.

The resulting map allowed the authors to characterize several regulatory modules, which broadly confirmed previous knowledge on the functional roles of the studied RBPs, but also pinpointed additional ones for some RBPs. For instance, the integrated regulatory map suggested that several RBPs can participate in different stages of RNA processing or, interestingly, play a role in the regulation of transcription. The authors selected 4 of these RBPs and performed additional experiments (e.g., RNA-seq, Ribo-Seq, ATAC-seq and ChIP-qPCR) to confirm the functional roles inferred from the IRIM.

Overall, the authors have generated an impressive amount of data that provide very interesting insights for the characterization and the understanding of post-transcriptional regulatory networks. This work represents a valuable resource for the community.

Nevertheless, I have several concerns to raise on different aspects of this work that should be addressed.

Comment 1.1

First of all, I believe that the authors should clarify the conceptual biology framework of their work.

For instance, reading the manuscript, it was not clear to me what the authors' definition of RNA regulon.

Previous established literature on this topic, which was partially overlooked by the authors in the introduction (e.g., PMID: 17572691 PMID: 18959479 PMID: 25436546), define an RNA regulon as an ensemble of functionally-related transcripts whose fate is coordinately regulated by RBPs and/or other regulatory molecules such microRNAs.

In the introduction, the authors underline that the simplistic model of “one RBP – one function – one regulon” is not sufficient to capture the complexity of post-transcriptional regulation (statement to which I agree), and suggest that a “one RBP – many functions – many regulons” is more appropriate.

To support this, they mention the example of MBNL1 that regulates splicing in the nucleus while modulating stability and localization of target RNAs in the cytoplasm. However, this does not tell us whether MBNL1 regulating the same mRNAs both in the nucleus and the cytoplasm or whether these mRNAs involved in the same cellular function(s), which is essential for the definition of a RNA regulon.

Later in the manuscript, while dissecting TAF15 role in splicing and RNA stability, they point out that RBP is involved in three different, although overlapping, regulons: alternative splicing, differential translation and stability (I have additional comments on this point, see below). However, it is not clear whether the regulated transcripts are involved in similar or different cellular functions. It seems to me that, in this case, the authors defined the regulon based on the role of the RBP rather than grounding it on the function(s) of the targets.

Response to 1.1

We thank the reviewer for this comment and the opportunity to clarify our conceptual framework. We define an RNA regulon as a group of transcripts that are bound and regulated as a unit by the same regulatory factors, such as RBPs, analogous to the concept of a DNA regulon. While it is common for co-regulated transcripts to be enriched for functional annotations, functional relatedness is not a necessary condition for being part of a regulon. In our revised manuscript, we explicitly define an RNA regulon as “a group of transcripts that are bound and regulated as a unit by the same regulatory factors,” and we cite the relevant literature (PMID: 17572691, PMID: 18959479, PMID: 25436546) on the subject that the reviewer kindly pointed out.

We also would like to thank the reviewer for the opportunity to further clarify the roles of TAF15 in splicing, translation, and stability (Fig. S8G and lines 308-320 in the revised manuscript). Fig. R1 demonstrates the number of significant GO terms from GO enrichment analysis for the genes that exhibit significant changes in translation, stability, or splicing upon TAF15 knockdown. As the reviewer has pointed out, our understanding of the regulon is based on the functional outcome of the RBP-RNA interactions. GO term enrichment analysis yields both a number of common terms shared across regulons (i.e. RNA processing and translation), and some specific terms which differ between regulons governed by TAF15. The latter reflects the aforementioned tendency of genes in the same regulon to be functionally related.

Figure R.1. Venn diagram of TAF15 RNA regulons. Shown are the numbers of significantly enriched GO terms (FDR < 0.05) for genes that exhibit significant changes in splicing, stability, or translation upon TAF15 knockdown, as captured by RNA-seq, Ribo-seq, and RNA-seq with α -amanitin, respectively. Manually selected representative GO terms are shown for each part of the diagram. Results of one-sided Fisher's exact test for each pairwise intersection are shown next to the corresponding area.

Comment 1.2

This brings me to another general concern regarding the definition of multifunctionality that the authors employ throughout the manuscript.

Multifunctional proteins, known also with a stricter definition as moonlighting proteins, are proteins that perform multiple distinct functions in the cell (e.g., PMID: 10087914, PMID: 20144902, PMID: 22696112, PMID: 26220711). Many RBPs can be defined as such (see aconitase for example), but they are often non-conventional RBPs (e.g.: PMID: 20554447, PMID: 26520658).

The authors seem to define multifunctional RBPs as they participate in different steps of RNA fate either in the nucleus and/or the cytoplasm. Given the established definition of multifunctionality, this should not be the case since they are always acting as RNA binders in different steps of the same process. Therefore, authors' definition looks to me closer to a "functional pleiotropy" of RBPs (i.e. doing the same task in different context) rather than protein multifunctionality or protein moonlighting.

Response to 1.2

We thank the reviewer for this comment; we have clarified the statements in the revised manuscript. We agree that “functional pleiotropy” might be the more appropriate term here. We have revised the manuscript accordingly (e.g. lines 177, 182, 202, etc).

Comment 1.3

When referring to their BioID dataset, the authors often mention “physical interactions” or “protein interactions”. However, as correctly described in the abstract, BioID proximity labelling allows to define *in vivo* protein neighborhoods of the proteins of interest and does not allow to infer any physical contact among these proteins. The authors should be consistent with this terminology in the manuscript.

Response to 1.3

We thank the reviewer for this comment. We made sure to make the use of terminology in the text consistent.

Comment 1.4

Overall, these concepts and definitions should be clarified, in particular in the introduction and discussion, and rephrased throughout the manuscript when needed.

Response to 1.4

We thank the reviewer for this comment. We have revised these parts of the paper to simplify and clarify the definitions.

Comment 1.5

In the discussion, understandably, the authors underline the relevance of their work and what are the next steps to take for improving our knowledge of post-transcriptional regulatory networks. Authors should also discuss any limitations that their study may have (e.g., number of studied RBPs, availability of relevant data such eCLIP, etc.).

Response to 1.5

We thank the reviewer for this comment. We have covered the limitations of the study in the updated discussion (lines 448-466 in the updated manuscript). Briefly, a key limitation of our proximity labeling methodology is the transgene expression of fusion proteins, which could potentially alter protein expression, localization, and function. Although our spot checks did not reveal significant localization changes, overexpression of fusion proteins could lead to artifacts in protein interaction data. Additionally, the Perturb-seq assay's limited cell sampling per perturbation might impact the breadth of data representation. Despite this, our statistical analyses have shown robustness even with added noise. Lastly, our high-throughput approach primarily serves as a foundation for hypothesis generation, necessitating future experimental validation and extension to unravel the complex dynamics of RBP-mediated regulation (as we have performed ourselves for select RBPs and regulons in this manuscript).

Besides these broad concerns, I have some more specific comments and suggestions:

Comment 1.6

a) The authors should consider mentioning in the introduction previous works that aimed at the characterization of post-transcriptional regulatory networks either computationally (e.g., PMID: 30867517 and PMID: 31733516) or through experimental approaches (e.g. PMID: 34133714, PMID: 37070168). Contrast with these previous works could be also beneficial for enriching the discussion section.

In particular, the dataset from Lang et al. could be used to support the IRIM clustering (e.g., are RBPs in the same clusters also are in physical contact?).

This analysis could be extended to additional resources, like the Human Reference Interactome (HuRI, <http://www.interactome-atlas.org/>), the Human Protein Complex Map (hu.MAP 2.0, <http://humap2.proteincomplexes.org/>), which includes BioPlex, and also to, and maybe more importantly, the Ribonucleoprotein Complex Map (rna.MAP, <http://rna.proteincomplexes.org/>) that collect human protein complexes containing RNAs.

The comparison to these protein interaction datasets could be beneficial not only for assessing the IRIM, but also to provide orthogonal “physical” support to the BioID data.

All this can be done in the same statistical framework that the authors used in the manuscript.

Response to 1.6

We thank the reviewer for this valuable suggestion. We have extracted the lists of interacting proteins from the datasets listed by the reviewer and compared them side by side with IRIM (Fig. R.2). As the Human Protein Complex Map (hu.MAP) and Zanzoni et al. ¹ covered most of the RBPs present in IRIM, we have decided to reserve them as “positive control” datasets for benchmarking. This helped us set up the proper thresholds for IRIM interactions to control false positives. Permutation tests revealed a statistically significant overlap between interactions detected in our study and these datasets (FDR = 0.01 for Human Map and 0.0015 for Zanzoni et al., using a 0.25 quantile as the integrated distance threshold for IRIM). These analyses have been included in the revised manuscript (Fig. 2F and Fig. 3SA).

The dataset published by Quattrone et al ² is a valuable resource that focuses on “RBP chains”, where an RBP of interest interacts with mRNA that encodes the target RBP, therefore controlling its expression. Given the difference in the nature of the interactions, we chose not to integrate this dataset with IRIM. Three other datasets, namely Human Reference Interactome, Lang et al ³ the Ribonucleoprotein Complex Map, unfortunately, provided only very sparse interaction maps (Fig. R.2), which do not allow for proper integration with IRIM.

Figure R.2. Comparisons of IRIM versus interaction maps from publicly available data. Upper triangle: the heatmap of IRIM as in Fig. 2A. Lower triangle: the heatmap showing the pairwise distances between RBPs inferred from the respective dataset. The pairs of RBPs where the interaction score is not available in the respective data source are shown in gray. The same

regulatory modules as in Fig. 2A are highlighted in red and yellow. The datasets shown are: (A) the Human Protein Complex Map, (B) Human Reference Interactome, (C) Lang et al ³ (D) Quattrone et al ², (E) the Ribonucleoprotein Complex Map (F) Zanzoni et al ¹.

Comment 1.7

b) The IRIM was generated through late multimodal data integration and hard clustering based on cosine distance to identify putative regulatory modules. The authors summarize the proposed approach in Figure S1. This figure should be improved by provide detailed explanation of the terms used in the equations.

Response to 1.7

We thank the reviewer for this suggestion. We have clarified the equation terms in the updated figure.

Figure S1. Multi-modal integration of RBP interaction data

Top: the three data modalities are shown on top; the datasets generated in this paper are highlighted with solid line, and the data downloaded from publicly available resources is highlighted with dashed line.

Middle: each dataset was preprocessed into a table, where the columns are RBPs and the rows are gene targets (shown in color for individual datasets). Every RBP was represented by a numeric column vector. The gene targets correspond to: for Perturb-Seq - individual genes, for BioID2 - protein binding partners, for eCLIP - mRNA binding targets. Difference between two numeric row vectors is shown on the right in the form of a histogram.

Bottom: the formulas applied at the key steps of the integration procedure are shown. (1): numeric column vectors were normalized by applying z-score transformation. (2): For each dataset, the cosine distances between pairs of individual RBPs were calculated. (3): The

resulting distances were then transformed into empirical P-values reflecting assay-specific inter-RBPs distances. (4): Finally, a single interaction score was measured for each RBP pair by combining the P-values from the three assays using logit aggregation.

Comment 1.8

In addition, it is not clear to me how the modules were functionally annotated. Was it done by labelling each cluster with the most frequent annotation among the RBPs or through enrichment analysis or consensus knowledge? Are these Gene Ontology annotations? Are the authors taking into account the nature of the evidence supporting the association between a given annotation and an RBP (i.e., including IEA annotations)? Please, explain better this point as this is key for the rationale of the work presented in the manuscript.

Response to 1.8

We thank the reviewer for this comment. We have included a detailed explanation of the annotation procedure in the Methods section of the revised manuscript. Modules were constructed for each RBP by taking its partners at IRIM distance < 25% quantile. Next, for each module we used BioID-derived GSEA results with NES > 0 for all the proteins included in the module, aggregating GO term *P*-values with logit method followed by FDR correction for the number of terms. All the GO terms with the aggregated significance FDR < 5% or with FDR passing 5% for at least 1 included RBP were kept and ranked by the former value to bring the most consistent GO pathways on top.

In summary, the functional annotation of the modules is primarily based on the aggregation of pathways enriched in proximity labeling data. The collection of BioID2 proximity label samples in triplicates, along with matched controls for each RBP (also in triplicates), allowed us to gather quantitative enrichment data, which enables detailed pathway analysis. By aggregating this pathway analysis and using strict statistical thresholds, we can confidently assign meaningful Gene Ontology terms to each module. This approach not only captures the most frequent annotations but also accounts for the strength and consistency of evidence supporting these annotations, thereby enhancing the reliability of our functional insights.

Comment 1.9

Related to this, in the lower panels of Figure 2, the authors present some illustrative examples of identified functional partners based on their integrated or dataset-specific cosine distance. These panels highlight that close RBP also shared a common function with the RBP interest. How many cases like these are found in the IRIM? It would be important to provide swarm plots for all the RBP in the IRIM as supplementary figure by highlighting those RBP with common function and see whether and how this correlates with distance.

Response to 1.9

We thank the reviewer for this comment. We tested whether RBPs share the common function with their closest neighbors and we observed that 63 out of 87 RBPs annotated in GO (72%) share their function with their closest partners (Fig. R.3, and Fig. S4 in the revised manuscript). We considered the GO terms that are enriched (NES score < -0.5, since we are interested in lower distance values) among the close neighbors of each query RBP. We then intersected these terms with the GO terms that contain the query protein. In 72% of cases, one or more GO terms belonging to the query RBP were also enriched among its closest binders.

Figure R.3. Swarm Plots for RBP Partners of the 90 RBPs. Each swarm plot represents the ordering of neighboring RBPs for a query RBP. Each point represents an individual RBP. The points are organized by the integrated distance from the specified RBP to the query RBP. Points highlighted in red share the query GO term annotation. The red-circled red dots are a part of the

GO term that also includes the query RBP (63 out of 87 RBPs annotated in GO, 72%, 3 RBPs are unannotated); the black-circled red dots represent the GO terms that do not include the query RBP (28% of the annotated RBPs). The query RBP and the query GO term are shown on the left.

Comment 1.10

In Figure S2D, the authors compared the IRIM with an analogous matrix built using STRING functional interactions. Below the heatmap is reported the color scale for STRING confidence score, but in the legend the authors refer to a distance metric. Please, clarify.

Response to 1.10

We thank the reviewer for this comment. We have clarified this issue in the revised manuscript. Shown are the STRING interaction confidence scores (multiplied by 1000).

Comment 1.11

Given the hard clustering method used by authors, RBPs in the IRIM could belong to only one cluster (“one RBP – one function – one regulon”). To spot RBPs that could participate in different functions, the authors indeed looked for off-diagonal signals. Did the authors try to apply different “soft” clustering approaches, like fuzzy clustering, to check whether some RBPs could be multi-clustered? I am asking this just for sake of comparison.

Response to 1.11

We thank the reviewer for this comment. We have performed fuzzy clustering of IRIM (Fig. R4, and Fig. S3C in the revised manuscript); the fuzzy clustering indicated 5 groups of RBPs assigned to multiple clusters (functionally pleiotropic); 2 of these clusters have been previously highlighted in Fig. 2. The fuzzy clustering provided additional insights by identifying RBPs that participate in multiple functions, which were not fully captured by the hard clustering approach. Specifically, it revealed RBPs with roles in both splicing and translation, demonstrating their functional pleiotropy. This additional layer of analysis highlights the complexity of RBP functions and supports the notion that some RBPs are involved in diverse regulatory pathways. The inclusion of this analysis has thus enriched our understanding and interpretation of the RBP regulatory modules, thanks to the reviewer's valuable suggestion.

Figure R.4. Fuzzy clustering of IRIM. Rows represent RBPs, columns represent clusters. The functionally pleiotropic groups of RBPs are highlighted in yellow. RBP groups from Fig. 2A are highlighted with solid frames, other groups are highlighted with dashed frames. For the clustering, we used the c-means algorithm with the degree of fuzzification set to 1.25, 10 clusters, and Manhattan distance.

Comment 1.12

c) The BioID proximity data has been also used by the authors to go beyond the RBP-RBP relationships and exploit the protein neighborhoods to gain further functional insights. As mentioned earlier in my report, I would perform an extensive comparison with additional protein interaction datasets to add a physical contact layer to the identified neighborhoods.

Response to 1.12

We thank the reviewer for this comment. We have performed comparisons of full IRIM, as well as IRIM with held-out datasets, with the Human Protein Complex Map and Zanzoni et al. ¹, along with STRING and OpenCell datasets (Fig. R5, and Fig. 3A in the revised manuscript). The other datasets, such as Human Reference Interactome, Lang et al ³, and the Ribonucleoprotein Complex Map, unfortunately, did not contain enough information of RBP-RBP interactions to perform proper dataset integration. Concerning the Human Protein Complex Map and Zanzoni et al., we have observed significant overlap in identified RBP-RBP interaction with IRIM (FDR = 0.01 and 0.0015, respectively).

Figure R.5. Boxplots and bubbles representing fractions of the interactions confirmed by the external databases among the RBP pairs with the inter-RBP distance lower than a certain quantile. Boxplots: 10^4 random shuffles of the IRIM; bubbles: the real data. The distances, left-to-right: the integrated distances from Fig. 2A, the distances from eCLIP and Perturb-seq integration, BioID and Perturb-seq integration, and BioID and eCLIP integration (see Methods). The external databases, top-to-bottom: STRING, OpenCell, OpenCell w/o STRING support, hu.MAP, hu.MAP w/o STRING support, Zanzoni et al., Zanzoni et al. w/o STRING support. Color fill denotes the external database used to calculate the fraction; size and line color denote the empirical P-value calculated from 10,000 shuffling iterations and FDR-corrected for the number of tests performed with different distance quantile thresholds.

Comment 1.13

I think that this dataset is not fully exploited and the authors could perform additional scrutiny. For instance, among RBP neighboring proteins, how many are known RBPs, how many are

non-conventional RBP? Do RBPs share neighbors? Is there any particular class of proteins that is present among all, or in some neighborhoods?

Response to 1.13

We thank the reviewer for this comment. As suggested, we have performed an additional analysis of BioID2 dataset (Fig. R6). First, we have calculated the fraction of canonical RBPs (those with RNA binding domains), non-canonical RBPs (those with no RNA binding domains that still bind RNA), and other proteins among the neighboring proteins (Fig. R6A), using the annotations from ⁴. This way we have observed a large number of non-RBP partners among the top 100 proximity proteins. Second, we have calculated the representation of different Pfam families ⁵ among the RBP neighborhood proteins (Fig. R6B); we have observed that several Pfam families, such as “RNA recognition motif”, “SET domain”, “helicase conserved C-terminal domain”, “WD40 domain”, are highly present among the top proximity partners across the variety of bait RBPs. Finally, we have analyzed the fraction of neighbors shared by RBPs (Fig. R6C); we have observed that, as expected, such analysis results in the clustering of RBPs similar to that observed in the heatmap showing the pairwise distances between RBPs based on the BioID2 dataset (Fig. S2C). These observations recapitulate the patterns that have been highlighted by the IRIM map.

Figure R.6. (A) Heatmap showing the fractions of canonical RBPs (those with RNA binding domains), non-canonical RBPs (those with no RNA binding domains that still bind RNA) and

unannotated proteins among the top 100 neighboring proteins for each query RBP. **(B)** Heatmap showing the number of proteins representing the most widely present Pfam families among the top 500 neighboring proteins for each query RBP. **(C)** Heatmap showing the number of partners shared by pairs of RBPs among their top 500 neighboring proteins.

Comment 1.14

In this part, the authors use in several occasions the “RBP-pathway” term when describing the functions associated to RBPs and/or regulatory modules. However, they use Gene Ontology terms from the different branches (BP, MF and CC) that are not necessarily “biological pathways”. Please, pick a more appropriate term.

Response to 1.14

We thank the reviewer for this comment. We would like to clarify that we used only BP GO terms (Biological Processes) for the analysis presented in the main figures. The other branches of GO terms (MF and CC) were used solely in the benchmarking analysis presented in Fig. S6E (Fig. S3E previously). We have clarified the terms accordingly.

Comment 1.15

The BioID data should be provided in a more reader-friendly in a tabular format with bait-prey pairs that can be considered as “bona fide” associations.

Likewise, for the results of GSEA (S12, S13, S14), a tabular format visualization of the data should be provided for easy exploration (i.e., RBP-annotation pairs, ES, NES, nominal p-value, FDR, $-\log_{10}(\text{FDR})$, list of neighboring proteins associated to the given annotation).

Response to 1.15

We thank the reviewer for this comment. As recommended in an earlier comment, we have submitted the lists of bona-fide BioID protein pairs to IMEx ⁶ (identifier IM-30059). We have also updated the results of GSEA to the tabular format (Data files S9, S10, S11).

Additionally, we have implemented a Web tool, RBP Browser (<https://goodarzilab.shinyapps.io/RBP-Browser/>), for the exploration of enriched pathways. RBP Browser offers an interactive map of human RNA-binding protein interactions, allowing users to query their RBP of interest and understand how it fits into the functional network of RNA regulation in human cells.

Comment 1.16

Out of curiosity, did the authors tested also for significantly depleted annotations in their GSEA analysis? This could provide some interesting insights on which functions are not in the proximity of a given RBPs.

Response to 1.16

We thank the reviewer for this comment. We have observed that the GO pathways that are significantly depleted in the proximity labeling profiles are largely the same across the queried RBPs. The two groups of pathways observed consistently across the queried RBPs are associated with nucleosomes and with biotin metabolism (Fig. R.7). The depletion of biotin metabolic genes is likely an artifact of the proximity labeling method, as this method involves comparing mass spectrometry profiles of biotin-treated cells with mass spectrometry profiles of cells grown in absence of biotin.

Regarding the depletion of nucleosome-related factors, this reflects the spatial organization within the cell. Cytoplasmic RBPs are separated from nuclear nucleosome-related factors. Additionally, even nuclear RBPs are found at sites of active transcription where nucleosomes are typically depleted⁷. This dual spatial separation leads to the observed depletion, highlighting that the proximity data we collected reflects the known aspects of RNA and chromatin biology.

Figure R.7. Examples of GO terms significantly depleted in the proximity labeling profiles, as reported by iPAGE⁸. Log fold change values were calculated for biotin-positive versus negative samples. These differences are partitioned into 15 discrete proximity bins. Bins to the left contain proteins that are abundantly found in the negative samples, whereas the ones to the right contain proteins found in biotin-positive samples. In the heat map representation, rows correspond to pathways and columns to consecutive proximity bins. Red entries indicate the enrichment of pathway genes in a given proximity bin. Enrichment and depletion are measured using hypergeometric p-values (log-transformed).

Comment 1.17

Regarding Figure 3B, the red over blue is not legible, even more for the asterisks indicating the relatively few significantly enriched BP annotations. Suggestion: try with complementary colors or a combination that could be clearly more distinguishable for color-blind people.

Response to 1.17

We thank the reviewer for this comment. We have updated the color scheme in Fig. 3B.

Comment 1.18

Finally, I strongly recommend the authors to submit their BioID protein proximity data, or at least the subset that they consider as “detected protein neighborhoods” of the bait RBPs, to one of the IMEx consortium databases (for instructions, check <http://www.imexconsortium.org/submit-your-data/>).

Response to 1.18

We thank the reviewer for this comment. We have submitted the BioID protein proximity data to IMEx ⁶ (identifier IM-30059).

Comment 1.19

The authors focused their attention on a handful of RBPs (ZCH3H11A, TAF15, QKI and ZNF800) for further validations of their functional roles inferred from the IRIM. In particular, they sought to verify the role of ZC3H11A and TAF15 in splicing regulation. For both RBPs there are some evidences of their “putative” involvement in such process. Especially for TAF15, previous works, cited also by the authors, proved a role in splicing, although in a mouse brain model. Indeed, in this manuscript the number of ASEs identified upon TAF15 knock-down (i.e., 190) is comparable to the one detected in Kapeli et al. (i.e. 187). Despite the fact that the experimental models are different, can the authors observe any overlap between the two experiments? In any case, the outcome of the TAF15 ASE experiment is confirmatory. Therefore, this should be reported as such throughout the manuscript.

It would be interesting as well to have some descriptive details on the different ASEs detected (intron retention, exon skipping, etc.) for both RBPs and their relative proportions.

Response to 1.19

We thank the reviewer for this comment. We agree that cross-validating the results of TAF15-KO splicing analysis is an important validation. Unfortunately, the data provided Kapeli et al ⁹ is low resolution and does not present a complete picture of TAF15-KO-induced splicing changes in human neurons. The main effort of Kapeli et al. was directed at the analysis of mouse splicing data, and the splicing changes occurring in the human neural progenitors were measured as a validation dataset and only at a low resolution (less than 2 million short reads). We have reanalyzed this data in the same way as the data collected in our work; MISO only detected 3224 skipped exon (SE) events, compared to 8237 in our data (3X less); the events of the other

types were detected at an even lower frequency. Only 17 of these SE events were identified as significantly changing upon TAF15 knockdown (Fig. R.8). Out of these 17 events, 2 were also identified as significantly changing in our data, showing statistically significant enrichment ($P = 0.038$, odds ratio = 7.2). To conclude, while the TAF15 splicing analysis in human cells from Kapeli et al. is of low resolution, we did observe a significant enrichment in the SE events that were identified as TAF15-dependent in both datasets.

Figure R.8. Vienn diagram showing the overlap of skipped exon (SE) events detected in control cells (upper) and significantly changing upon TAF15 KD (lower) in our study compared to those detected in Kapeli et al ⁹.

Comment 1.20

In Figure 5E, the authors show the overlap among the set of genes whose transcripts are potentially regulated by TAF15 at three different steps of the RNA fate process. The statistical significance of these overlap is assessed with a one-sided Fisher's exact test. It seems that only two pair sets show a significant overlap. However, what I get from the figure is that when testing the overlap, the authors forgot to include the 13 genes that are present in all three sets. Is this the case? If yes, please redo the test, otherwise clarify this in the figure legend and/or

manuscript. Please also provide exact p-value from the Fisher's test for all the pairs and adjust them for multiple testing correction.

Response to 1.20

We thank the reviewer for this comment. The 13 genes were included in the analysis but this was not communicated clearly in our figure. We have updated the diagram in the revised manuscript (Fig. 5E). We have also updated the P-values using multiple testing correction (Benjamini–Hochberg FDR).

Comment 1.21

The PCA plots in Figures S4C and S5E show that two replicates (TAF15-KD and WT-untreated, respectively) are not relatively close to each other. Can the authors comment on that?

Response to 1.21

We thank the reviewer for this comment. In Fig. S8E (Fig. S5E previously), the two WT-untreated samples are indeed far from each other. As the two replicates were grown in two separate plates, it is possible that the gene expression profile was affected by cell growth conditions, such as confluency. Nevertheless, these two samples are located closer to each other than either of them is to the corresponding TAF15-KD sample, indicating that the replicate-dependent changes in the transcriptomic profile were smaller than those caused by the introduction of TAF15-KD.

As for Fig. S7D (Fig. S4C previously), the two replicates of TAF15 knockdown were created independently by transducing the K562-CRISPRi cells with TAF15 sgRNA. The difference in transcriptome profile could arise from the lentiviral integration in different genomic locations across the cell population. To make sure both replicates were displaying efficient TAF15 knockdown, we compared the TAF15 expression in each RNA-seq sample (measured by counts-per-million, CPM), and we observed efficient knockdowns in each replicate (Fig. R.9, included in the manuscript as Fig. S7C). This indicates that despite the difference in transcriptomic profiles, both replicates can be used as representative cell populations of the TAF15 knockdown condition.

Figure R.9. Relative expression (in CPM) of TAF15 and ZC3H11A in the RNA-seq samples related to Fig. S7D.

Comment 1.22

Finally, I strongly recommend the authors to provide, beside the raw data deposited in GEO, the processed analyzed data (ASEs, Ribo-Seq, ATAC-seq, etc.) as supplementary material or as an archive in data sharing repositories such as Zenodo along with a proper documentation.

Response to 1.22

We thank the reviewer for this comment. We have deposited the processed data on Zenodo (identifier 11556393) (DOI: ¹⁰)

Comment 1.23

e) I downloaded and tested the code associated with the manuscript. The documentation is sufficient for running the provided scripts. Nevertheless, the authors should provide more detailed comments within the R scripts. In addition, the script should also take care of installing automatically the required libraries if they are missing on the user operating system (OS). However, as the authors correctly state, the installation of the libraries may depend on the OS

present on the user computer. As a matter of fact, I was not able to install some libraries and run the scripts because I do not have a recent version of R on my computer. Thus, the optimal solution is to provide the analysis script in a dockerized environment with all the dependencies installed.

Response to 1.23

We thank the reviewer for this comment. We improved the technical usability of our scripts by providing a premade Conda environment file containing all the required packages. In addition, we updated the instructions to make the installation process seamless.

Additional comments:

Comment 1.24

* I suggest the authors to organize the methods' section in a way that it is consistent with the flow of the manuscript in the result section.

Response to 1.24

We thank the reviewer for this comment. We have reorganized the methods accordingly.

Comment 1.25

* The supplementary tables are poorly described and annotated. Please, provide more details descriptions within the data files. Readers should be able to understand what the tables contains without going back and forth between this material and the manuscript.

Response to 1.25

We thank the reviewer for this comment. We have revised and improved the annotations of the supplementary tables in the revised manuscript.

Comment 1.26

* The STRING database provide information about protein functional interactions that are not necessarily based only on physical interaction data. Sometime the authors forget to mention this detail in the manuscript.

Response to 1.26

We thank the reviewer for this comment. We have clarified in the manuscript that the interactions listed in STRING include direct (physical) and indirect (functional) associations.

Comment 1.27

Reviewer #1 (Remarks on code availability):

I downloaded and tested the code associated with the manuscript. The documentation is sufficient for running the provided scripts. Nevertheless, the authors should provide more detailed comments within the R scripts. In addition, the script should also take care of installing automatically the required libraries if they are missing on the user operating system (OS). However, as the authors correctly state, the installation of the libraries may depend on the OS present on the user computer. As a matter of fact, I was not able to install some libraries and run the scripts because I do not have a recent version of R on my computer. Thus, the optimal solution is to provide the analysis script in a dockerized environment with all the dependencies installed.

Response to 1.27

Answered above

Reviewer #2:

General comments

- What are the noteworthy results?

Compared to TF interactions that regulate gene expression, the post-transcriptional RBP interactions are still not quite clear. In this study, the authors constructed a map of RBP-RBP interactions by integrating data from physical and functional RBP interactions, and highlighted the value of this RBP interaction map in revealing the functions of some RBPs.

Comment 2.1

- Will the work be of significance to the field and related fields? How does it compare to the established literature? If the work is not original, please provide relevant references.

Although most of the important references on this topic have been cited, they still missed some critical ones – such as Xiao et al, Cell (PMID 31251911) and Li et al, Genome Biology (PMID 28886744).

Response to 2.1

We thank the reviewer for this comment. We have revised the introduction and the discussion sections, making sure to cite the relevant papers suggested by the reviewer.

Comment 2.2

It is also important to show the overlap between the RBP modules predicted by this study and those revealed by previous studies.

Response to 2.2

We thank the reviewer for this comment. First, we updated the benchmarking of IRIM (Fig. R.10A; Fig. 2E,F in the revised manuscript) to reflect the intersections of IRIM with the key existing datasets such as OpenCell, STRING, and hu.MAP. To this end, we compared the resilience of the different database arrangements by gradually introducing random noise and comparing the distances from a given RBP to its closest or farthest neighbor. We observed a similar change in the structures of IRIM and the STRING database. We have also estimated the significance of the IRIM intersection with other databases by non-parametric permutation-based tests (Fig. R.10B). We observed statistically significant overlap between IRIM interactions and these databases (FDR = 0.031 for STRING, 0.00017 for OpenCell, 0.12 for BioPlex, and 0.01 for hu.MAP using a 0.25 quantile as the integrated distance threshold). At this threshold, we identified 1001 RBP-RBP pairs, with 776 of these interactions being novel (not reported in the STRING database), and an average of 22 contacts per RBP, a five-fold increase in interactions compared to STRING (see Supplementary Data File 14). Overall, we observe that IRIM's structure is equally resilient to noise as one of STRING-DB, and that IRIM's identified interactions overlap significantly with those reported by the other databases. Additionally, we have included direct comparisons of IRIM with several other external databases, as suggested by Reviewer 1 (Fig. R.2). However, in most cases the external datasets were too sparse on the given set of RBPs to perform an informative comparison.

Figure R.10: Assessment of Downsampling on IRIM.

(A) Rearrangements in RBP matrices: This panel demonstrates the alterations in the structure of the Integrated Regulatory Interaction Map matrix due to random shuffling, depicting changes in distance to the closest and farthest partner RBP. Downsampling was conducted by shuffling distance values of varying fractions of RBPs (0% to 100%). **(B)** Percent of RBP pairs passing IRIM distance < 25% quantile that intersect STRING, OpenCell, Hu.Map, and Zanzoni et al. Violin and boxplots are based on 10^4 random shuffling iterations; red dots represent the percent of the real IRIM distances.

Comment 2.3

- Does the work support the conclusions and claims, or is additional evidence needed?

The authors integrated three types of data – proximity-based, target-based and function-based data – to predict the RBP modules. Given that the RBPs can interact with each other at both the levels of transcription and post-transcription, the results from this study cannot distinguish whether the RBP modules may function at which level. Their BioID experiments can capture both RNA-bound and chromatin-bound RBPs; the data from the Pertub-seq experiments should be treated as an integrated readout of RBP perturbation from both the levels of transcription and post-transcription; while the eCLIP data can only support RBP interactions at the RNA level.

The authors claim that the RBP modules identified by the integrated analysis are “post-transcriptional regulatory modules” (Figure 2). I suggest the authors double check the CHIP-seq data from Xiao et al, Cell (PMID 31251911) to exclude the false positive ones or analyze the RNA-associated and chromatin-associated RBP modules separately.

Response to 2.3

We thank the reviewer for this comment. We filtered out all RBP-gene interactions identified in CHIP-Seq data for 26 RBPs from K562 cells. Specifically, we checked whether the protein binds to the gene body or promoter and then filtered out the corresponding RBP-RNA interactions in eCLIP data. This arguably harsh filtering led to a 6.6% reduction in the number of module RNA targets in total, as shown in the updated Supplementary Data File 15. We agree with the reviewer that it is challenging to completely distinguish between transcriptional and post-transcriptional interactions. However, given that eCLIP data, which unlike CHIP-seq captures direct physical interactions between RBPs and RNA, was a foundation of our analysis and since only 6.6% of the interactions were filtered out based on CHIP-Seq data, the vast majority of the identified associations (and therefore modules) are likely to be post-transcriptional. Therefore, our integrated analysis primarily reflects post-transcriptional regulatory modules.

Comment 2.4

- Are there any flaws in the data analysis, interpretation and conclusions? Do these prohibit publication or require revision?

I would suggest the authors to analyze the motif component of the RBP modules. I'm wondering whether the RBPs with the same or similar motif component or not would be in the same RBP module.

Response to 2.4

We thank the reviewer for this comment. Indeed, we observed that the RBPs co-participating in the same modules tend to have higher motif similarity than the RBPs coming from different modules. To test this, we have collected the position weight matrices (PWM) for the available RBPs (26 RBPs in total) across two motif databases, namely oRNAmotif¹¹ and CISBP-RNA¹². We then calculated the similarities between the pairs of motifs using MACRO-APE¹³, and classified the pairs of RBPs into two groups, “similar motifs” or “different motifs”, based on the similarity values. To then test whether the motifs are more similar for members of the same module versus the RBPs from different modules, we performed Fisher's exact test and observed a P-value of 0.044. This supports the hypothesis that RBPs that participate in the same module might also share the motif component.

Comment 2.5

- Is the methodology sound? Does the work meet the expected standards in your field?

In the main figure, the authors should show the overlap of their RBP pairs detected from BioID and predicted RBP modules with the data from some popular PPI databases. The authors put these results in the supplementary files, but I think these results are critical for evaluating the confidence of the predicted RBP modules.

Response to 2.5

We thank the reviewer for this comment. We have updated the main (Fig. 2E,F in the revised manuscript) and supplementary (Fig. S3A,B in the revised manuscript) figures to highlight the comparison of RBP interactions identified by IRIM versus the ones identified in the external datasets.

Comment 2.6

- Is there enough detail provided in the methods for the work to be reproduced?

The Methods section was well written and easy to follow.

Some writing issues – for example, “i.e.” should be “i.e.,” (Line 147).

Response to 2.6

We thank the reviewer for this comment. We have fixed the errors pointed out.

Reviewer #2 (Remarks on code availability):

The authors provided a good Readme file, which is helpful for running the codes.

Reviewer #3:

Comment 3.1

This study used a systems biology approach to identify the post-transcriptional regulatory modules between RNA-binding proteins (RBPs) and target regulons. A comprehensive dataset has been generated in cell lines using RNA-seq, CRISPRi knockdown, BioID2-RBP fusion cell line, mass spectrometry, Perturb-seq, ribosome profiling, ATAC-seq, and ChIP-qPCR etc. Together with public data sources like CLIP ENCODE, post-transcriptional regulatory modules

(as shown in Fig. 2) were identified as clusters in a heatmap. Functional annotation of RBPs was also conducted based on BioID-mediated proximity labeling. Specific examples such as ZC3H11A and TAF15 were also presented to illustrate the capability of RBPs regulating multiple independent regulons via distinct pathways. Data analytics were described with necessary details given (e.g., tool names and version numbers). Overall, this study presented a solid piece of work on unveiling regulatory relationships between RBPs and target regulons, with novel scientific insights gained by digging into selected examples. There are a few minor concerns/suggestions though:

1. The choice of analytic pipeline tools was not clearly justified. At least some of the tools used are not of the state-of-the-art or not of the best performance (e.g, RNA-seq analysis tools). One sentence may be added to give some justification.

Response to 3.1

We thank the reviewer for this comment. To our knowledge, we have used the tools considered state-of-the-art. MISO is a robust and reliable tool that has been widely used and validated in many recent publications¹⁴⁻¹⁷. While MISO does not use CPU and RAM resources as efficiently as rMATS, SUPPA2 and other tools¹⁸, it is known for its robust performance in identifying the alternative splicing events¹⁷. We chose it for its proven consistency it offers in RNA-seq analysis (lines 1183-1184 in the revised manuscript). In the other analysis, to our knowledge, we have also used the state of the art tools, such as the Kundaje lab ATAC-seq pipeline used by ENCODE¹⁹, MaxQuant²⁰ for the mass spectrometry data analysis, and SCVI²¹ and SCANPY²² for Perturb-Seq analysis.

Comment 3.2

2. Instead of heatmap as well as biclustering algorithm, the relationship between RBP units and target regulons might be better depicted using graphs/networks. In that case, the modules obtained will correspond to graph/network communities, and the interactions between RBP units and regulons are thus the interactions between communities (a multiple-to-multiple relationship).

Response to 3.2

We appreciate the reviewer's suggestion to employ a network/graph-based visualization approach. As shown in Fig. R.11 (Fig. S3D in the revised manuscript), we have adopted this technique to depict the relationships between RBP units as graph communities, which intuitively clusters major RNA metabolic processes such as ribosome assembly and mitochondrial RNA metabolism. This method highlights the interactions within and between these communities effectively.

Following Reviewer 1's recommendation, we also implemented fuzzy clustering, as depicted in Fig. R.4 (Fig. S3C in the revised manuscript). This approach was particularly useful for identifying RBPs that participate in multiple modules, corroborating the community structures observed in the graph-based visualization and providing continuity with the patterns initially observed in the heatmap of Fig. 2.

Finally, demonstrating the interactions of RBP modules with their extensive target RNAs poses significant technical challenges, primarily due to the sheer volume and RBP-RNA interactions involved (thousands of interactions per RBP module).

Overall, the consistency of RBP groupings across these visualizations reinforces the robustness of our findings.

Figure R.11: Graph visualization of IRIM

Vertices represent individual RBPs; edges depict the pairs of RBPs with an integrated distance less than 0.05. The communities of RBPs, related to different groups of RNA processes, are colored and highlighted as in Fig. 2.

Comment 3.3

3. A clearer definition of RBP unit is expected for readers to understand the results. For instance, is it completely data-driven or a mixture of data and knowledge?

Response to 3.3

We thank the reviewer for this comment and the opportunity to clarify our framework. In the revised manuscript, we explicitly define the regulatory modules, that we have annotated in a data-driven way: “regulatory modules – which we define as a set of RBPs that share significant functional interactions (Suppl. Data 16)” (lines 133-134 in the revised manuscript). We provide a comprehensive annotation of regulatory modules in Suppl. Data 16.

Reviewer #3 (Remarks on code availability):

Source codes were provided for data integration and BioID annotation. The rest of computing codes for, e.g., RNA-seq analysis, was not provided, which is acceptable as standard scripts are publicly available at many places.

References

1. Zanzoni, A., Spinelli, L., Ribeiro, D. M., Tartaglia, G. G. & Brun, C. Post-transcriptional regulatory patterns revealed by protein-RNA interactions. *Sci. Rep.* **9**, 4302 (2019).
2. Quattrone, A. & Dassi, E. The Architecture of the Human RNA-Binding Protein Regulatory Network. *iScience* **21**, 706–719 (2019).
3. Lang, B. *et al.* Matrix-screening reveals a vast potential for direct protein-protein interactions among RNA binding proteins. *Nucleic Acids Res.* **49**, 6702–6721 (2021).
4. Beckmann, B. M. *et al.* The RNA-binding proteomes from yeast to man harbour conserved enigmRBPs. *Nat. Commun.* **6**, 10127 (2015).
5. Mistry, J. *et al.* Pfam: The protein families database in 2021. *Nucleic Acids Res.* **49**, D412–D419 (2021).
6. Del Toro, N. *et al.* The IntAct database: efficient access to fine-grained molecular interaction data. *Nucleic Acids Res.* **50**, D648–D653 (2022).
7. Lee, C.-K., Shibata, Y., Rao, B., Strahl, B. D. & Lieb, J. D. Evidence for nucleosome

- depletion at active regulatory regions genome-wide. *Nat. Genet.* **36**, 900–905 (2004).
8. Goodarzi, H., Elemento, O. & Tavazoie, S. Revealing global regulatory perturbations across human cancers. *Mol. Cell* **36**, 900–911 (2009).
 9. Kapeli, K. *et al.* Distinct and shared functions of ALS-associated proteins TDP-43, FUS and TAF15 revealed by multisystem analyses. *Nat. Commun.* **7**, 12143 (2016).
 10. Creators Khoroshkin, Matvei1 Buyan, Andrey Show affiliations 1. University of California, San Francisco. 'Systematic Identification of Post-Transcriptional Regulatory Modules': *Preprocessed Data*. doi:10.5281/zenodo.11556393.
 11. Benoit Bouvrette, L. P., Bovaird, S., Blanchette, M. & Lécuyer, E. oRNAmont: a database of putative RNA binding protein target sites in the transcriptomes of model species. *Nucleic Acids Res.* **48**, D166–D173 (2020).
 12. Ray, D. *et al.* A compendium of RNA-binding motifs for decoding gene regulation. *Nature* **499**, 172–177 (2013).
 13. Vorontsov, I. E., Kulakovskiy, I. V. & Makeev, V. J. Jaccard index based similarity measure to compare transcription factor binding site models. *Algorithms Mol. Biol.* **8**, 23 (2013).
 14. Gabel, A. M. *et al.* Multiplexed screening reveals how cancer-specific alternative polyadenylation shapes tumor growth in vivo. *Nat. Commun.* **15**, 959 (2024).
 15. Shi, L. *et al.* SULT1A1-dependent sulfonation of alkylators is a lineage-dependent vulnerability of liver cancers. *Nat Cancer* **4**, 365–381 (2023).
 16. Jalloh, B. *et al.* The Drosophila Nab2 RNA binding protein inhibits m6A methylation and male-specific splicing of Sex lethal transcript in female neuronal tissue. *Elife* **12**, (2023).
 17. Olofsson, D., Preußner, M., Kowar, A., Heyd, F. & Neumann, A. One pipeline to predict them all? On the prediction of alternative splicing from RNA-Seq data. *Biochem. Biophys. Res. Commun.* **653**, 31–37 (2023).
 18. Muller, I. B. *et al.* Computational comparison of common event-based differential splicing tools: practical considerations for laboratory researchers. *BMC Bioinformatics* **22**, 347

(2021).

19. Kundaje lab ATAC-seq pipeline. <https://www.encodeproject.org/pipelines/ENCPL792NWO/>.
20. Cox, J. *et al.* Accurate proteome-wide label-free quantification by delayed normalization and maximal peptide ratio extraction, termed MaxLFQ. *Mol. Cell. Proteomics* **13**, 2513–2526 (2014).
21. Lopez, R., Regier, J., Cole, M. B., Jordan, M. I. & Yosef, N. Deep generative modeling for single-cell transcriptomics. *Nat. Methods* **15**, 1053–1058 (2018).
22. Wolf, F. A., Angerer, P. & Theis, F. J. SCANPY: large-scale single-cell gene expression data analysis. *Genome Biol.* **19**, 15 (2018).

Reviewer #1 (Remarks to the Author):

The authors have addressed my concerns in a satisfactory manner.

Reviewer #2 (Remarks to the Author):

The authors have performed more analyses in the revised version and addressed my comments. I recommend acceptance.

Reviewer #3 (Remarks to the Author):

The authors have addressed all my concerns. While there exist some better tools for, e.g., RNA-seq alignment, which were not used in this work's pipeline, I deemed it as a minor drawback. The revision looks acceptable to me for publication.